# Targeting the Epigenetic Non-Coding RNA MALAT1/Wnt Signaling Axis as a Therapeutic Approach to Suppress Stemness and Metastasis in Hepatocellular Carcinoma

**DOI:** 10.3390/cells9041020

**Published:** 2020-04-20

**Authors:** Hang-Lung Chang, Oluwaseun Adebayo Bamodu, Jiann-Ruey Ong, Wei-Hwa Lee, Chi-Tai Yeh, Jo-Ting Tsai

**Affiliations:** 1Department of General Surgery, En Chu Kong Hospital, New Taipei City 237, Taiwan; changhl0321@gmail.com; 2Department of Health Care Management, Yuanpei University of Medical Technology, Hsinchu 300, Taiwan; 3Department of Hematology and Oncology, Cancer Center, Taipei Medical University-Shuang Ho Hospital, New Taipei City 235, Taiwan; dr_bamodu@yahoo.com (O.A.B.); ctyeh@s.tmu.edu.tw (C.-T.Y.); 4Department of Medical Research and Education, Taipei Medical University-Shuang Ho Hospital, New Taipei City 235, Taiwan; 5Department of Emergency Medicine, School of Medicine, Taipei Medical University, Taipei 110, Taiwan; 12642@s.tmu.edu.tw; 6Department of Emergency Medicine, Shuang-Ho Hospital-Taipei Medical University, New Taipei City 235, Taiwan; 7Department of Pathology, Taipei Medical University-Shuang Ho Hospital, New Taipei City 235, Taiwan; whlpath97616@s.tmu.edu.tw; 8Department of Medical Laboratory Science and Biotechnology, Yuanpei University of Medical Technology, Hsinchu 300, Taiwan; 9Department of Radiology, School of Medicine, College of Medicine, Taipei Medical University, Taipei 110, Taiwan; 10Department of Radiation Oncology, Taipei Medical University-Shuang Ho Hospital, New Taipei City 235, Taiwan; 11Graduate Institute of Clinical Medicine, College of Medicine, Taipei Medical University, Taipei City 110, Taiwan

**Keywords:** LncRNA MALAT1, HCC-SCs, tumorspheres, drug resistance, cancer recurrence

## Abstract

Background: With recorded under-performance of current standard therapeutic strategies as highlighted by high rates of post-treatment (resection or local ablation) recurrence, resistance to chemotherapy, poor overall survival, and an increasing global incidence, hepatocellular carcinoma (HCC) constitutes a medical challenge. Accumulating evidence implicates the presence of HCC stem cells (HCC-SCs) in HCC development, drug-resistance, recurrence, and progression. Therefore, treatment strategies targeting both HCC-SCs and non-CSCs are essential. Methods: Recently, there has been an increasing suggestion of MALAT1 oncogenic activity in HCC; however, its role in HCC stemness remains unexplored. Herein, we investigated the probable role of MALAT1 in the SCs-like phenotype of HCC and explored likely molecular mechanisms by which MALAT1 modulates HCC-SCs-like and metastatic phenotypes. Results: We showed that relative to normal, cirrhotic, or dysplastic liver conditions, MALAT1 was aberrantly expressed in HCC, similar to its overexpression in Huh7, Mahlavu, and SK-Hep1 HCC cells lines, compared to the normal liver cell line THLE-2. We also demonstrated a positive correlation between MALAT1 expression and poor cell differentiation status in HCC using RNAscope. Interestingly, we demonstrated that shRNA-mediated silencing of MALAT1 concomitantly downregulated the expression levels of β-catenin, Stat3, c-Myc, CK19, vimentin, and Twist1 proteins, inhibited HCC oncogenicity, and significantly suppressed the HCC-SCs-related dye-effluxing potential of HCC cells and reduced their ALDH-1 activity, partially due to inhibited MALAT1-β-catenin interaction. Additionally, using TOP/FOP (TCL/LEF-Firefly luciferase) Flash, RT-PCR, and western blot assays, we showed that silencing MALAT1 downregulates β-catenin expression, dysregulates the canonical Wnt signaling pathway, and consequently attenuates HCC tumorsphere formation efficiency, with concurrent reduction in CD133+ and CD90+ HCC cell population, and inhibits tumor growth in SK-Hep1-bearing mice. **Conclusions:** Taken together, our data indicate that MALAT1/Wnt is a targetable molecular candidate, and the therapeutic targeting of MALAT1/Wnt may constitute a novel promising anticancer strategy for HCC treatment.

## 1. Background

Long non-coding RNAs (lncRNAs) are >200 nucleotide long transcripts thought to be unable to code for any protein and are often characterized by evolutionarily conserved micro-homologies, secondary structure, and functions, with unique expression in differentiated tissues or specific cancer types [1,2,3]. As genome-wide analyses continue to reveal an increasingly broad landscape of functionally mutated non-coding transcriptomes, there is accruing evidence of the ‘driver’ role of the expression and/or activity of several lncRNAs in malignant transformation and oncogenic phenotypes [4]. This is consistent with common knowledge that lncRNAs, acting as cues for specific cellular states or bioactivities [5], may help identify cellular dysfunctions or pathologies such as malignancies, predict tumor behavior, provide prognostic insight, or inform anticancer therapeutic strategies [4,5].

Metastasis-associated lung adenocarcinoma transcript 1 (MALAT1), acting as a regulator of transcription for many genes, including some involved in the cell cycle, cell migration, and metastasis, is overexpressed in cancerous tissues and is associated with the proliferation and metastasis of malignant cells, such as in lung cancer [6] and HCC [7,8] cells. Recently, it was suggested that increased expression of MALAT1 increased the proportion of pancreatic cancer stem cells (CSCs), decreased their sensitivity to anticancer drugs, enhanced their self-renewal capacity, accelerated their tumor angiogenesis, and promoted tumor growth [9], while the downregulation of MALAT1 promoted the proliferation of the glioma stem cell line SHG139S by suppressing the expression of stemness markers, Nestin and Sox2 [10]; however, the role of MALAT1 in the induction and/or maintenance of the CSCs-like phenotype in HCC remains unexplored, and there is a dearth of data detailing the role of MALAT1 in the maintenance and proliferation of CSCs in HCC.

Liver CSCs, referred to herein as HCC-SCs, and delineated by stemness markers such as cluster of differentiation (CD)133, CD90, CD44, CD47, CD13, CD24, delta-like non-canonical notch ligand 1 (DLK1), epithelial cell adhesion molecule (EpCAM), leucine-rich repeat containing G protein-coupled receptor 5 (Lgr5), intercellular adhesion molecule 1 (ICAM-1), or keratin 19 protein, are characterized by an enhanced capacity to self-renew, initiate tumors, metastasize, and evade therapy; thus, they are implicated in the therapy-resistance, metastasis, high recurrence, and poor survival rates of patients with HCC [11]. Therefore, eradication of HCC-SCs is touted as a probable effective approach to improve the outcome of HCC patients. We hypothesized that MALAT1 plays a crucial role in the biology of HCC-SCs, is molecularly targetable, and that direct targeting of the HCC-SCs may constitute an efficacious therapeutic strategy for HCC treatment.

Therefore, in the present study, we explored the probable role(s) and underlying molecular mechanisms of MALAT1 in HCC-SCs by statistical analyses of HCC-related big data and the functional probe of the effect of short-hairpin RNA (shRNA)-mediated downregulation of MALAT1 expression using the human HCC cell lines, Huh7, Mahlavu, SK-Hep1, and hepatoblastoma cells HepG2, as well as HCC tumor xenograft mice models. Based on our findings, we believe that the therapeutic targeting of MALAT1 is a promising novel therapeutic approach to treat HCC.

## 2. Materials and Methods

This study was approved by the Institutional Human Research Ethics Review Board (TMU-JIRB No. 201302016) of Taipei Medical University and performed in accordance with the Institutional Guide for the Care and Use of Laboratory Animals of the Taipei Medical University (Approval number: LAC-2018-0572).

### 2.1. Drugs and Reagents

The compound cisplatin (CDDP, #479306; ≥99.9% purity) was purchased from Sigma Aldrich Co. (St. Louis, MO, USA). Stock solution of 1 mM dissolved in dimethyl sulfoxide (DMSO, #W387520; Sigma-Aldrich Co., St. Louis, MO, USA) was stored at −20 °C until use. CDDP was dissolved in DMSO and diluted further to desired concentration in sterile complete culture medium immediately prior to use. Propidium iodide (PI, #81845), phosphate buffered saline (PBS, #P7059), sulforhodamine B (SRB, #230162) dye, TRIS base (#93352), and acetic acid (#A6283) were all purchased from Sigma (Sigma-Aldrich Co., St. Louis, MO, USA), Gibco^®^ Dulbecco’s modified Eagle medium (DMEM, #11960077), fetal bovine serum (FBS, #A31605), and trypsin-EDTA (#25300054) were purchased from Thermo Fisher (Thermo Fisher Scientific Inc., Waltham, MA, USA). The QIAGEN OneStep RT-PCR kit (#210212) was obtained from Qiagen (QIAGEN Inc., Germantown, MD, USA). BD Pharmingen™ FITC Annexin-V apoptosis detection kit I (#556547) was from BD (BD Biosciences, San Jose, CA, USA).

### 2.2. Cell Culture

The human HCC cell lines, Huh7, Mahlavu, SK-Hep1 and hepatoblastoma cells HepG2, as well as the non-tumor liver cell line THLE-2 were purchased from ATCC and were cultured in DMEM or RPMI1640 supplemented with 10% (*v*/*v*) heat-inactivated fetal bovine serum (FBS), Penicillin (100 IU/mL) and Streptomycin (100 μg/mL), in humidified 5% CO2 atmosphere at 37 °C. Cells were sub-cultured when they attained a confluence ≥90, and the media were changed every 48–72 h. Establishment and growth of primary HCC culture cells were performed strictly according to the protocol presented by Cheung PF et al. [12].

### 2.3. MALAT1 Silencing

The shRNA specifically targeting MALAT1 was constructed by CRISPR Gene Targeting Core Lab (Taipei Medical University, Taiwan). For MALAT1 silencing, SK-Hep1 and HepG2 cells cultured to ~80% confluence were transfected with MALAT1 shRNA (shRNA#1: Forward 5′-CCGGAAAGCCCTGA ACTATCACACTCTCGAGAGTGTGATAGTTCAGGGCTTTTTTG-3′; Reverse 5′-AATTCAAAAAAAAGCCCTGAACTATCACACTCTCGAGAGTGT GATAGTTCAGGGCTTT-3′ or shRNA#2: Forward 5′-CCGGAATCTGTAAGCAGTTT GTATGCTCGAGCATACAAA CTGCTTACAGATTTTTTTG-3′; Reverse 5′-AATTCAAAAAAA TCTGTAAGCAGTTTGTATGCT CGAGCATACAAACTGCTTACAGATT-3′) or scrambled shRNA (scrMALAT1, 5′-GACCTGTACGC-CAACACAGTG-3′) as negative control (Appendix A). Transfection was performed using Lipofectamine 2000 Transfection reagent (#11668019, Thermo Fisher Scientific Inc., Carlsbad, CA, USA) according to the manufacturer’s protocol. The stably transfected shMALAT SK-Hep1 and HepG2 cells were selected with 1μg/mL puromycin (#A11138-03, GIBCO, Grand Island, NY, USA) and harvested 48 h post-transfection.

### 2.4. Western Blot Analysis

After washing cells with PBS twice and lysing with ice-cold RIPA lysis buffer (#20-188, Sigma-Aldrich Co., St. Louis, MO, USA), the total protein lysate from the wild-type or shMALAT1 HCC cells was centrifuged and pellet collected. Protein concentration was then quantified using Pierce™ BCA protein assay kit (#23227, Thermo Fisher Scientific Inc., Waltham, MA, USA). Equal amount of protein lysate from each sample was run on 10% sodium dodecyl sulfate polyacrylamide gel electrophoresis (SDS-PAGE) then protein transferred to Polyvinylidene difluoride (PVDF) membranes which were blocked in 1X PBS containing 5% skimmed milk, and then incubated with the specific primary antibodies against β-catenin (1:1000, β-Catenin (6B3) Rabbit mAb #9582P, Sigma), CD133 (1:1000, #MAB4310, Sigma), ALDH-1 (1:1000, (D4R9V) Rabbit mAb #12035, Sigma), c-Myc (1:1000, c-Myc Antibody (9E10) (sc-40), Sigma), CK19 (1:1000, monoclonal antibody SAB3300019, Sigma), Stat3 (1:1000, Stat3 (79D7) Rabbit mAb #4904, Sigma), vimentin (1:1000, Anti-Vimentin antibody (ab137321), Sigma), Twist1 (1:1000, Twist (Twist2C1a) Antibody sc-81417, Sigma), cyclin D1 (1:1000, Cyclin D1 Antibody (A-12) (sc-8396), Sigma), Axin2 (1:1000, Axin2 Antibody SAB3500619, Sigma), LEF1 (1:1000, LEF1 (C12A5) Rabbit mAb #2230, Sigma), or DKK1 (1:1000, DKK1 Antibody #4687, Sigma) at 4 °C overnight for protein detection in Appendix A. After overnight probing, protein bands were identified using anti-mouse or anti-rabbit horseradish peroxidase (HRP)-linked secondary antibody at room temperature for 1 h. The signal was detected using UVP^®^ imaging system (Analytik Jena US LLC., Upland, CA, USA). β-actin (1:10000, 8H10D10, Mouse mAb #3700) was used as loading control. The gray value was quantified and analyzed using Image J software. The experiment was repeated 3 times.

### 2.5. Immunohistochemistry (IHC) Analysis

All human tissues were obtained from surgical resection specimens of HCC patients at Taipei Medical University-Shuang-Ho Hospital (New Taipei City, Taiwan). Tissue microarray (TMA) slides were established, then heat-based antigen retrieval was performed in EDTA-containing buffer, sections blocked with 5% bovine serum albumin (BSA)/1% HISS/0.1% Tween20 solution, and incubated with primary recombinant antibody against MALAT1 (1:400 dilution; #MOB-4044z, Creative Biolabs, NY, USA) overnight, at 4 °C. MALAT1 immunoreactivity/positivity was detected using the mouse IgGκ light chain binding protein conjugated to horseradish peroxidase m-IgGβ BP-HRP (#sc-516102; Santa Cruz Biotechnology, Inc., Santa Cruz, CA, USA) and the EXPOSE mouse and rabbit specific HRP/DAB detection IHC kit (#ab80436, Abcam plc., Cambridge, MA, USA). This study was approved by the Institutional Human Research Ethics Review Board (TMU-JIRB No. 201302016) of Taipei Medical University.

### 2.6. RNA Isolation and Quantitative RT-PCR

Total RNA was isolated using the highly efficient genomic DNA removal RNeasy plus kit (#74136, Qiagen, Valencia, CA, USA), according to the manufacturer’s protocol. The NanoDrop ND1000 spectrophotometer (Nyxor Biotech, Paris, France) was used in determining total RNA concentration. Reverse transcription of 1 μg total RNA with 2 μg random hexamers (GE Healthcare, Amersham, Buckinghamshire, UK) and Invitrogen™ SuperScript™ III reverse transcriptase (#18080-044; Thermo Fisher Scientific Inc., Grand Island, NY, USA) according to manufacturer’s instructions. The following primers were used in this study: MALAT1 forward 5′-AAAGCAAGGTCTCCCCACAAG-3′, reverse 5′-GGTCTGTGCTAGATCAAAAGGC-3′; c-Myc forward 5′-GGCTCCTGGCAAAAGGTCA-3′, reverse 5′-CTGCGTAGTTGTGCTGATGT-3′; cyclin D1 forward 5′-GCTGCGAAGTGGAAACCATC-3′, reverse 5′-CCTCCTTCTGCACACATTTGAA-3′; Axin2 forward 5′-ACAACAGCATTGTCTCCAAGCAGC-3′, reverse 5′-GCGCCTGGTCAAACATGATGGAAT-3′; LEF1 forward 5′-AGAACACCCCGATGACGGA-3′, reverse 5′-GGCATCATTATGTACCCGGAAT-3′; DKK1 forward 5′-CCTTGGATGGGTATTCCAGA-3′, reverse 5′-CCTGAGGCACAGTCTGATGA-3′; GAPDH (internal control) forward 5′-ATCATCCCTGCCTCTACTGG-3′, reverse 5′-GTCAGGTCCACCACTGACAC-3′. The data were displayed as 2^−ΔΔCt^ values and are representative of at least three independent experiments.

### 2.7. Matrigel Invasion Assay

For analysis of the invasion potential of scrambled shRNA or shMALAT1 HCC cells, the 24-well plate Transwell system was used. 3.5 × 10^4^ cells were seeded onto the 0.2% Matrigel -layered upper chambers of the inserts (BD Bioscience, 8 μm pore size) containing serum-free media, while the lower chambers contained media with 10% FBS serving as chemo-attractant. Media was discarded after 24 h incubation, and then non-invaded cells remaining on the upper side of the inserts were removed with sterile cotton swabs while invaded cells underneath the filter membranes were fixed with 3.7% formaldehyde for 1 h and stained with crystal violet. The invaded cells were visualized and evaluated under microscope.

### 2.8. Colony Formation Assay

Approximately 2 × 10^2^ scrambled shRNA or shMALAT1 SK-Hep1 and HepG2 cells seeded into a 6-well cell culture plate were incubated for 10–12 days at 37 °C in humidified 5% CO_2_ atmosphere. Cells were then washed twice with PBS, fixed in cold methanol, stained with 0.005% crystal violet, washed several times, and air-dried. The total number of colonies formed with diameter ≥100 μm in each well was estimated over five randomly selected visual fields in assays performed four times in triplicate. The visible colonies were manually counted, and the rate of colony formation was calculated with the following equation: (number of colonies/numbers of seeded cells) × 100%.

### 2.9. Tumorsphere Formation Assays

HCC tumorspheres were generated from scrambled shRNA or shMALAT1 SK-Hep1 and HepG2 single-cell suspension. SK-Hep1 or HepG2 cells were seeded at a density of 5 × 10^4^ per well into Corning^®^ Costar^®^ ultra-low attachment 6-well plates (Corning, NY, USA) containing growth factors for CSCs enrichment stem cell medium (Nutristem-XF, Biological Industries, Israel) and incubated at 37 °C in humidified 5% CO_2_ incubator for 5–7 days, followed by visualization of the primary tumorspheres (diameter ≥100 μm) under an inverted microscope. Secondary tumorspheres were generated by dissociating the primary tumorspheres by trypsinization, pipetting dissociated primary tumorspheres through a 22G needle (Thermo Fisher Scientific Inc., Bartlesville, OK, USA) to obtain single-cell suspension. After dissociation of the primary tumorspheres, cells were seeded as earlier described for primary tumorspheres. After 5–7 days of culture, secondary orospheres consisting of ≥100 µm were counted, and images taken under microscope. Tumorsphere size, quantity and formation efficiency were evaluated. Tumorsphere formation efficiency (TFE) was evaluated using the formula: TFE = (number of sphere formed/number of single cells plated) × 100.

### 2.10. RNAScope Analysis

RNAScope was performed strictly following a previously described protocol by Wang F., et al. [13].

### 2.11. RNA Immunoprecipitation

LncBase Predicted v.2 bioinformatics tools (http://carolina.imis.athena-innovation.gr/) was used to predict the potential interaction of MALAT1 and CTNNB1. RNA immunoprecipitation (RIP) analysis was performed using EZ-Magna RIP™ RNA-Binding Protein Immunoprecipitation Kit (#17-701; Sigma-Aldrich Corporation, St. Louis, MO, USA) strictly following the manufacturer’s recommended protocol. The RNA from the whole cell lysates and the RIP fractions were extracted with TRIzol according to the manufacturer’s instructions. The relative mRNA expression levels of MALAT1 and CTNNB1 were determined using RT-qPCR analysis, as described above. The relative mRNA enrichment in the RIP fractions was computed based on the ratio of relative mRNA levels in the RIP fractions and the relative mRNA levels in the whole cell lysates (input).

### 2.12. Fluorescence-Activated Cell Sorting (FACS) Analysis of ALDH Activity

The Aldelfluor™ Kit (#01700; STEMCELL Technologies, Interlab Co., Ltd. Taipei, Taiwan) was used to determine ALDH enzymatic activity in SK-Hep1 cancer cells. The SK-Hep1 cells were trypsinized, washed with PBS thrice, and counted using a hemocytometer. 1 × 10^6^ SK-Hep1 cells were then re-suspended per 1 mL of Aldefluor buffer containing ATP-binding cassette (ABC) transporter inhibitor to prevent Aldefluor™ dye efflux. The re-suspended HCC cells were then incubated in a dark room at 37 °C for 1 h, washed in Aldefluor™ buffer, and maintained constantly at 4 °C following the manufacturer’s instructions. High ALDH activity was evaluated using the FL1 channel of the BD FACSAria™ system and FACSDiva™ software v. 6.1.2 (BD Biosciences, San Jose, CA, USA); thereafter, the collected SK-Hep1 cells were analyzed for ALDH activity based on fluorescence intensity and side scatter with FACS, as well as low side scatter.

### 2.13. TOP/FOP Flash Luciferase Reporter Assays

Scrambled shRNA and shMALAT1 SK-Hep1 or HepG2 cells were transfected with TOP/FOPFlash reporter with plasmids (Millipore, Bedford, MA, USA) and GFP pZsGreen for 48 h, then they were harvested, lysed with Promega Luciferase Assay Kit (Promega, Madison, WI, USA) and luciferase activities evaluated on a Centro X53 LB 960 microplate luminometer (Berthold Technologies GmbH & Co. KG, Bad Wildbad, Germany). For all experiments, TOP/FOPFlash reporters with plasmids or shRNAs were transfected in quadruplets plus an expression plasmid for β-galactosidase (β-gal). The expression of β-gal determined on an ELISA reader was used to normalize luciferase activity for transfection efficiency. The luciferase values normalized by the transfection efficiency from TOPFlash and FOPFlash quadruplets were averaged and used in calculating the TOP/FOPFlash ratio.

### 2.14. Side Population Analysis

For the side population (SP) analysis, SK-Hep1 cells were washed with PBS three times and re-suspended DMEM supplemented with 10% FBS at 37 °C in humidified 5% CO2 incubator. After initial incubation for 10 min, the SK-Hep1 cells were labeled with 5 μg/mL Hoechst 33,342 dye for 1.5 h, and then counter-stained with 1 μg/mL propidium iodide (PI) to identify dead cells. This was followed by the analysis and sorting of 1 × 10^6^ viable HCC cells using a BD FACSAria™ II fluorescence-activated cell sorting (FACS) system. The Hoechst dye was excited at 355 nm and its fluorescence measured at two wavelengths using optical filters 450/20 nm band-pass filter for Hoechst blue and 635 nm long-pass edge filter for Hoechst red. PI labeling was measured through a 630/BP30 filter for identifying dead cells.

### 2.15. Mice Tumor Xenograft Studies

All tumor xenograft studies were approved by the institutional research ethics committee and the Institutional Animal Care and Use Committee (Approval number: LAC-2018-0572). 4–6-week-old female NOD/SCID mice were purchased from the BioLASCO (BioLASCO Taiwan Co., Ltd. Taipei, Taiwan). Mice were inoculated with 1 × 10^6^ shMALAT1#2 SK-Hep1 tumorsphere-derived cells (N = 5), while the control mice were inoculated with the scrambled control SK-Hep1 tumorsphere-derived cells (N = 5). Tumor growth was then monitored by in vivo bioluminescence imaging (IVIS200 imaging system, Caliper Life Sciences Inc., Hopkinton, MA, USA) for 5 weeks. Post-experiment, all animals were humanely sacrificed by tumor dislocation and the tumor samples were harvested for further analyses.

### 2.16. Statistical Analysis

All experiments were performed at least three times in triplicates, and the results expressed as the mean ± SD. The Shapiro-Wilk test was used for testing normality of our data. Differences between groups were analyzed using the unpaired 2-tailed Student’s t-test for parametric data, while for non-parametric data, we used the Mann-Whitney test. Statistical analysis was performed using the Statistical Package for the Social Sciences (IBM Corp. Released 2017. IBM SPSS Statistics for Windows, Version 25.0., Armonk, NY: IBM Corp.) and a *p*-value < 0.05 was considered statistically significant.

## 3. Results

### 3.1. The lncRNA MALAT1 Is Aberrantly Expressed in HCC Tissues and Cell Lines

To understand the biological significance of MALAT1 in human HCC, we evaluated the expression profile of MALAT1 in the Wurmbach liver database (n = 75) consisting of HCC (n = 35), liver cell dysplasia (n = 17), cirrhosis (n = 13) and normal liver (n = 10) samples, using the Oncomine platform (https://www.oncomine.org). We observed that the expression of MALAT1 was highest in HCC, compared to other pathological liver conditions or normal liver; compared to the normal liver samples, MALAT1 expression in the HCC and liver cell dysplasia samples was up-regulated by 3.23-fold (*p* = 1.38 × 10^−6^; *t*-test = 6.28), and 3.17-fold (*p* = 2.59 × 10^−6^; *t*-test = 6.28), respectively (Figure 1A). Similarly, using different human HCC cell lines, we demonstrated that compared to its expression in the epitheloid well-differentiated Huh7 cell (1.5-fold, *p* < 0.01), the expression of MALAT1 was profoundly enhanced in the fibroblastoid poorly-differentiated Mahlavu (1.7-fold, *p* < 0.01) and HepG2 human hepatoblastoma cell lines (2.6-fold, *p* < 0.001), SK-Hep1 (2.8-fold, *p* < 0.001) HCC cell lines [14], and markedly low expression in the normal liver THLE-2 cell line (0.2-fold, *p* < 0.001) (Figure 1B). Consistent with the above, results of our comparative analyses of paired HCC and adjacent non-tumor tissue samples (n = 8 pairs) from the Taipei Medical University-Shuang-Ho Hospital patients cohort (n = 72) using the quantitative PCR demonstrate that the expression of MALAT1 is enhanced in most (~75%) HCC samples compared to their non-tumor counterpart, with a mean expression which is 2.66-fold higher in the HCC in comparison to the non-tumor group (*p* < 0.01) (Figure 1C). These results indicate that increased MALAT1 expression is characteristic of fibroblastoid, highly malignant HCC cells and tissues, and suggest its involvement in the poor cellular differentiation status of HCC and its associated aggressive phenotype.

### 3.2. MALAT1 Expression in Liver Cancer Positively Correlates with Poor Cellular Differentiation Status and Disease Progression

To confirm the suggested probable involvement of MALAT1 in the poor cellular differentiation status and disease progression, we first analyzed the expression level of MALAT1 in a public cancer database. Using the University of California Santa Cruz (UCSC) Xena platform, we examined likely association or correlation between MALAT1 expression and the sample types, histological types, and histological grade (cellular differentiation status) of samples in the TCGA Liver cancer (LIHC) cohort (n = 438). We demonstrate that the larger proportion of MALAT1^high^ HCC cells were moderately differentiated (G2), poorly differentiated (G3), or undifferentiated (G4), while the well-differentiated (G1) cells were mostly MALAT1^low/null^ (Figure 2A). In addition, using RNAsope analyses of 3 differently-staged HCC cases from the Taipei Medical University-Shuang-Ho Hospital patients cohort (n = 72), we showed that in contrast to the lack of MALAT1 expression in adjacent non-tumor tissues, MALAT1-positive cells were widely distributed in HCC tissues, and per intensity, MALAT1 was strongly, moderately or mildly expressed in Stage III/IV (n = 42), II (n = 18) or I (n = 12) HCC tissues, respectively (Figure 2B and Appendix A). This is further supported by results of our analyses of primary, recurrent and non-tumor liver samples from the TCGA Liver cancer (LIHC) cohort (n = 438) which demonstrated that compared to its expression in the non-tumor/normal liver tissues, MALAT1 is significantly expressed in primary and recurrent liver cancer (1-way Anova: *p* = 2.40 × 10^−11^, F-value = 25.93) (Appendix A). Consistent with the above, statistically, RNAscope analyses of tissues from our local HCC cohort consisting of 36 pairs of HCC and adjacent non-tumor tissues revealed a 3.86-fold (*p* < 0.001) up-regulated MALAT1 expression in the HCC samples compared to the non-tumor group (Figure 2C). Furthermore, using the RNAscope staining assay, we demonstrated for the first time to the best of our knowledge, that compared to the non-tumor liver tissue with no expression of MALAT1 or liver cirrhosis and well-differentiated HCC tissues with mild expression of MALAT1, the moderately-to-poorly-differentiated HCC exhibited strong and well-distributed expression of MALAT1 (Figure 2D). With 70% of the well-differentiated HCC samples with low/no expression of MALAT1, and 65% of the poorly differentiated HCC cases with high MALAT1 expression, we demonstrated a 30–40% (*p* < 0.001) inter-grade differential MALAT1 expression (Figure 2E). These data are indicative of the aberrant MALAT1 expression in HCC progression and suggestive of its probable role as a biomarker of poor cellular differentiation in HCC.

### 3.3. MALAT1 Expression Modulates HCC Oncogenicity and Stemness via Interaction with Wnt/β-Catenin

Having demonstrated that MALAT1 expression is associated with the progression and recurrence of HCC earlier, against the background of increasing evidence linking cancer progression and recurrence to the presence of CSCs [15,16], we investigated if and how MALAT1 expression modulates the stem cell-like phenotype of HCC cells using the wild type SK-Hep1 culture cells and MALAT1 knockdown clones shMALAT1#1 and shMALAT1#2. By FACS, compared to 7.2% CSCs-like side population (SP) cells identified in control SK-Hep1 culture cells bearing scrambled shRNA, we observed only 1.8% and 1.1% CSCs-like SP cells in shMALAT1#1 and shMALAT1#2 SK-Hep1 cultures, respectively (Figure 3A, *upper*). Further, in similar assay, we evaluated the effect of MALAT1 expression on ALDH activity using the Aldefluor™ flow cytometry assay. Side scatter of incident light in control and shMALAT1 SK-Hep1 cells showed that silencing MALAT1 reduced the ALDH1 activity in the shMALAT1#1 and shMALAT1#2 (Figure 3A, *lower*). These results were validated by Western blot analyses results showing that compared to the scrambled control cells, the expression levels of CD133, ALDH1 and β-catenin proteins in shMALAT1 SK-Hep1 or HepG2 cells were significantly down-regulated (Figure 3B). We also observed significant reduction in the number of invaded shMALAT1#1 (75%, *p* < 0.001) or shMALAT1#2 (83%, *p* < 0.001) cells, compared to the scrambled control (Figure 3C, *left*); similarly, clonogenicity was suppressed in shMALAT1#1 and shMALAT1#2 SK-Hep1 cells, respectively, compared to the scrambled control (Figure 3C, *right*). In addition, we demonstrated that in MALAT1 knockdown HCC clones, the expression of c-Myc, CK19, β-catenin, p-Stat3, Stat3, vimentin and Twist1 proteins was significantly suppressed, compared to the scrambled control cells (Figure 3D). Observing a correlation in the expression of β-catenin, a principal component of the canonical Wnt signaling pathway, which is implicated in the expansion of CSCs population and induction of epithelial-to-mesenchymal transition (EMT) [17], we sought to understand if the demonstrated oncogenic and stemness activities of MALAT1 were via its direct interaction or interplay with β-catenin. Following sequence-based prediction of interaction between MALAT1 (NCBI Reference Sequence: NR_002819.4) and, β-catenin (NCBI Reference Sequence: NP_001895.1), with 0.85 and 0.97 interaction probabilities using random forest (RF) and support-vector machines classifiers, respectively, we further used the Schrödinger^®^ PyMOL molecular graphics software version 2.3.2 (https://pymol.org/2/) to create a visualization of the molecular interaction between MALAT1 (PDB ID: 4PLX) and β-catenin (PDB ID: 3BCT), thus showing that MALAT1 directly targets and binds to β-catenin with a root-mean-square deviation (RMSD) of 4.0 Å, docking score of 19478, β-catenin/MALAT1 complex interface area of 3077.50 Å^2,^ and an atomic contact energy (ACE) of −298.23 kcal/mol , and that this interaction may bear some functional connotations, especially in the context of HCC metastatic and CSCs-like phenotype (Figure 3E). For additional validation, we probed the effect of immunoprecipitating β-catenin with MALAT1 on the level bound transcripts using the RNA immunoprecipitation (RIP) assay, since this assay allows interrogation of the physical association between RNA binding proteins and transcripts of interest. Our RIP analysis demonstrates that both MALAT1-β-catenin molecules can be co-immunoprecipitated, but other interacting molecules could be involved (Figure 3F).

### 3.4. MALAT1 Oncogenic Activity in HCC Is Mediated by the Wnt/β-Catenin Signaling Pathway

Despite evidence that β-catenin is a principal component of the canonical Wnt signaling pathway and that transcriptional response to Wnt signal activation is coordinated by a complex mesh of factors that bind to β-catenin in the nucleus [18], the mechanisms underlying MALAT1-mediated oncogenicity and CSCs-like phenotypes in HCC cells remain uncoupled with the activation of Wnt/β-catenin signaling. Thus, using the TOP/FOP flash luciferase reporter assay, we investigated the effect of MALAT1 on β-catenin/TCF-dependent transcriptional activity as an indicator of the role of MALAT1 in the modulation of the canonical Wnt signaling pathway, and the probable role this may play in MALAT1-mediated oncogenicity. The TOP/FOP flash reporter activities in shMALAT1#2 SK-Hep1 and HepG2 cells were significantly inhibited (SK-Hep1: 71%, *p* < 0.01; HepG2, 52%, *p* < 0.01), compared to the scrambled control cells (Figure 4A). Furthermore, the mRNA expression of β-catenin and its downstream targets, which are effector genes of the Wnt/β-catenin pathway, namely cyclin D1, c-Myc, Axin2, LEF1, and DKK1 in SK-Hep1 and HepG2 cells were evaluated by quantitative PCR. The relative mRNA expression levels of cyclin D1, c-Myc, Axin2, LEF1, and DKK1 were all significantly down-regulated in the MALAT1-silenced SK-Hep1 and HepG2 cells compared to their scrambled control counterparts (Figure 4B). Consistent with the above, western blot analysis showed that the expression of β-catenin, Stat3, c-Myc, cyclin D1, Axin2, LEF1, and DKK1 proteins in the shMALAT1#2 SK-Hep1 and HepG2 cells were significantly reduced, compared to their control scrambled shRNA counterparts (Figure 4C,D). These results are indicative of the probable role of MALAT1 in the modulation of the Wnt/β-catenin signaling pathway via direct interaction with β-catenin in the oncogenicity of HCC.

### 3.5. Silencing of MALAT1 Is Associated with a Reduced CD133^high^CD90^high^ HCC Population with Suppressed HCC Tumorsphere Formation In Vitro

Having shown that the oncogenic activity of MALAT1 in HCC is associated with enhanced expression of pluripotency markers CD133 and ALDH1, and is mediated, at least in part, by the Wnt/β-catenin signaling pathway, in vitro, we further investigated the translational relevance of these findings in the context of current anticancer chemotherapy, and in in vivo murine HCC models. In corroboratory assays, we demonstrated that concomitantly with reduced tumorsphere size and quantity in the shMALAT1 cells, the immunoreactivity of the cancer-associated pluripotency marker CD133, and mesenchymal and/or liver stem cell marker CD90/THY-1 [19,20] was significantly suppressed in the shMALAT1 cells, compared to the scrambled control (Figure 5A). We also demonstrated that shMALAT1#1 and shMALAT1#2 markedly inhibited the self-renewal potential of the SK-Hep1 (shMALAT1#1: 82% inhibition, *p* < 0.001; shMALAT1#2: 92.3% inhibition, *p* < 0.001) or HepG2 (shMALAT1#1: 83% inhibition, *p* < 0.05; shMALAT1#2: 93.5% inhibition, *p* < 0.01) in primary and secondary tumorspheres (Figure 5B). Next, against the background that cisplatin (CDDP)-based hepatic arterial infusion chemotherapy enhances the objective response rate and survival benefit in patients with advanced HCC (19), we examined the probable effect of shMALAT1 on the response to CDDP treatment. Interestingly, we observed that contrary to conventional knowledge, CDDP increased the population of CD133^high^CD90^high^ SK-Hep1 (28.1%) or HepG2 (22.7%) cells, compared to vehicle-treated scrambled control (Figure 5C), suggesting a CDDP-induced HCC-SCs phenotype. However, this enhanced CD133/90 positivity was markedly repressed in the shMALAT1#1 and shMALAT1#2 HepG2 or SK-Hep1 cells. shMALAT1#2 in combination with CDDP significantly inhibited population of CD133^high^CD90^high^ across both HepG2 and SK-Hep1 cells (Figure 5C). Furthermore, to gain better insight into the relation between enhanced CD133/CD90 positivity and MALAT1 expression, our re-analysis of the TCGA-LIHC cohort (n = 374) showed that consistent with our earlier results, the expression of CD133/PROM1 is positively correlated with CD90/THY1 (r = 0.359, *p* = 8.55 × 10^−13^), and that MALAT1 exhibits positive correlation with CD133/PROM1 (r = 0.165, *p* = 1.33 × 10^−3^) and CD90/THY1 (r = 0.165, *p* = 1.33 × 10^−3^) (Figure 5D).

### 3.6. MALAT1 Is Required for Xenograft Tumorigenesis and Tumor Growth of HCC Cells In Vivo

Having established that MALAT1 is required for the growth of HCC cells in vitro, we decided to determine if MALAT1 was also required for in vivo growth of HCC cells. According to in vivo mice tumor xenograft models, we observed significant lower bioluminescent intensity in the mice bearing shMALAT1#2 tumors as compared to the control. For instance, the fold change in bioluminescence in the shMALAT1#2 group was approximately 198.5-fold (*p* < 0.01) lower than that in the control counterpart at week 5 (Figure 6A). As shown in Figure 6A, MALAT1-depleted xenografts grew much slower than controls. At the end of experiment (week 5), control tumors had grown to much larger sizes than that of MALAT1-depleted ones (Figure 6B). Moreover, MALAT1-depleted xenografts tumors displayed decreased Ki67 and PCNA expression, indicating these tumors were more proliferative inhibition than control tumors (Figure 6C). These data indicate a role for silencing MALAT1 in the suppression of HCC-SC-induced and/or maintained tumorigenicity and cancer stemness.

## 4. Discussion

While there has been considerable improvement in the diagnosis and treatment of HCC over the last decade, the incidence of HCC remains high, therapy response low, predisposition to developing resistance to contemporary chemo- and/or radio-therapy high, and treatment failure more common, as is evidenced by increasing incidence and mortality rates over the last 10 years, coupled with a relatively low ~50% 2-year survival rate and perplexing 10% 5-year survival rate in the United States [21]; Thus, managing HCC remains a medical enigma, requiring the discovery or identification of novel actionable molecular targets and development of a more effective therapeutic strategy.

Our evolving understanding of the intra- and inter-tumoral heterogeneity of the HCC bulk which is reminiscent of a constitutive stochastic responsiveness of individual HCC cells to known anticancer therapeutics and modality, portends the futility of focused targeting of proliferating non-CSCs and heralds the therapeutic efficacy of targeting and attenuating the growth or proliferation of the HCC-SCs pool to abrogate cancerous cell re-population, enhanced metastatic phenotype, and disease recurrence [11,15,20]; highlighting the need for a therapeutic strategy that effectively target and eliminate the CSCs in HCC. Exploiting our understanding of the critical role of aberrantly expressed MALAT1 and activated Wnt/β-catenin signaling in the oncogenicity and stemness activities of several solid tumor types [6,7,8,9,10,11], the present study summarily provides evidence that aggressive HCC cells are characteristically MALAT1high and that this positively correlated with an enhanced CD133^high^CD90^high^ HCC-SCs pool and associated with Wnt/β-catenin signaling-mediated marked up-regulation in HCC oncogenicity and pluripotency.

Our work showed that the lncRNA MALAT1 is overexpressed in liver cancer tissues and cell lines (Figure 1). This finding is corroborated by Malakar P., et al, who recently suggested a role for increased MALAT1 expression in the development of HCC through interaction with serine and arginine-rich splicing factor 1 and activation of mTOR signaling [22]. These findings are clinically relevant especially as accruing evidence indicate that the aberrant expression of MALAT1 positively correlates with bigger tumor size, disease progression, and poor clinical outcome, making MALAT1 a principal molecular candidate for new clinically translatable lncRNA-based anticancer therapeutic strategies [23].

We also showed that the LncRNA MALAT1 overexpression in liver cancer positively correlates with poor cellular differentiation status and disease progression (Figure 2). This is consistent with findings indicating that the siRNA-mediated genetic ablation or antisense oligonucleotides (ASOs)-mediated systemic silencing of MALAT1 in the mouse mammary tumor virus (MMTV) - polyoma middle tumor-antigen (PyMT) mice breast cancer models concomitantly attenuated tumor growth and elicited significant differentiation into cystic tumors, enhanced cell adhesion, with reduced migration and metastasis [24]. Thus, our findings associating MALAT1 expression with poor cellular differentiation status and disease progression are of translational significance considering the implication of poorly differentiated cancerous cells in increased metastasis incidence and invariable poor clinical outcome of HCC [25].

Furthermore, we demonstrated that MALAT1 expression modulates HCC oncogenicity and ALDH/CD133-associated stemness via interaction with β-catenin (Figure 3). Hepatic CSCs are a small sub-population of undifferentiated cancerous liver cells, implicated as drivers of cancer initiation, metastasis, resistance to therapy, and disease recurrence; these HCC-SCs pool is enriched for and isolated based on the cell functions attributable mainly to side population (SP), enhanced ALDH activity and auto-fluorescence, as well as immunophenotypes defined by the cell surface CD13, CD24, CD44, CD47, CD90, CD133, DLK1, EpCAM, or ICAM-1 proteins [26], therefore the our demonstrated ability of MALAT1 to modulate the expression of ALDH and CD133 with their associated stemness phenotypes is not only relevant in understanding the pathobiology of HCC, but also therapeutically significant. The present finding is consistent with a previous finding by our team indicating that MALAT1 overexpression with correlated aberration in KDM5B/hsa-miR-448 ratio promotes breast cancer aggressiveness and stemness [27], as well as with others indicating MALAT1 promotes stemness in esophageal squamous cell carcinoma [28] and gastric cancer [29].

Moreover, we showed for the first time to the best of our knowledge, that MALAT1 oncogenic activity in HCC is mediated by direct interaction with β-catenin and activation of the Wnt/β-catenin signaling pathway (Figure 3 and Figure 4). This is consistent with evidence showing that the therapeutic inhibition of MALAT1 suppressed the invasion and metastasis of colorectal cancer cells via mediation of the Wnt/β-catenin signaling pathway [30], and that up-regulated MALAT1 in high-fat food fed ApoE−/− mice enhanced the nuclear translocation of β-catenin and activation of Wnt/β-catenin signaling, with associated increase in angiogenesis biomarker CD31, endothelial cell marker von Willebrand factor (vWF), α-SMA, and vimentin, consequently leading to enhanced endothelial-to-mesenchymal transition (EndMT) [31].

In addition, we presented evidence showing that silencing MALAT1 is associated with reduced CD133^high^CD90^high^ HCC population with suppressed HCC tumorsphere formation (Figure 5) and tumor growth in vivo (Figure 6). This is particularly relevant in the context of the role of CSCs in therapeutic response and disease course and prognosis. In fact, it was recently shown that aberrantly expressed CD90 and CD133 are strongly associated with reduced sensitivity to anticancer therapy, and contribute to disease progression and poor survival rates (overall survival and disease-free survival) in patients with hepatoblastoma [32], which is consistent with the demonstrated therapeutic success of targeting the CD90/β3 integrin/AMPK/CD133 signaling axis in liver cancer reported by Chen WC et al. [20], and concordant with current knowledge of the pluripotent/stemness-defining role of CD133 and CD90 in HCC [25].

## 5. Conclusions

In conclusion, as we depicted in our schematic abstract (Figure 7), the present study provides evidence that the the molecular targeting of the lncRNA MALAT1 effectively depletes the CD133^high^CD90^high^ HCC pool, inhibits the constitutive stemness of selected CD133^high^CD90^high^ HCC cells, impairs the nuclear translocation of β-catenin, quells the aberrant activation of the Wnt/β-catenin, represses cisplatin-induced HCC-SCs enrichment, abrogates cancerous liver cell metastasis and clonogenicity, as well as suppresses in vivo tumor initiation and growth.

## Figures and Tables

**Figure 1 cells-09-01020-f001:**
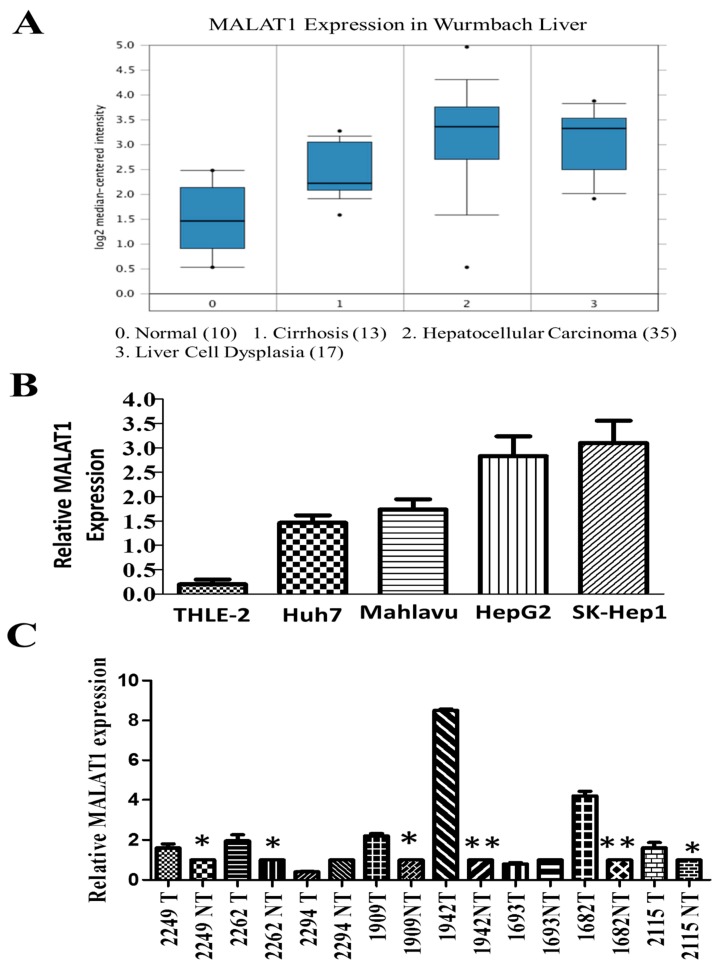
LncRNA MALAT1 is over-expressed in liver cancer tissues and cell lines. (**A**) Differential MALAT1 expression in HCC (n = 35, median 3.352) followed by liver cell dysplasia (n = 17, median 3.32), cirrhosis (n = 13, median 2.21) and normal liver (n = 10, median 1.456) from analyses of the Oncomine Wurmbach liver dataset. 0 = Normal; 1 = cirrhosis; 2 = hepatocellular carcinoma; 3 = liver cell dysplasia. (**B**) Graphical representation of relative MALAT1 mRNA expression levels in normal liver cell line THLE-2, HCC Huh7, Mahlavu, SK-Hep1 and HepG2 human hepatoblastoma cell lines. U6 served as internal control. (**C**) Comparative analyses in paired clinical HCC and non-tumor liver samples using quantitative PCR method. * *p* < 0.05, ** *p* < 0.01; T, tumor; NT, non-tumor.

**Figure 2 cells-09-01020-f002:**
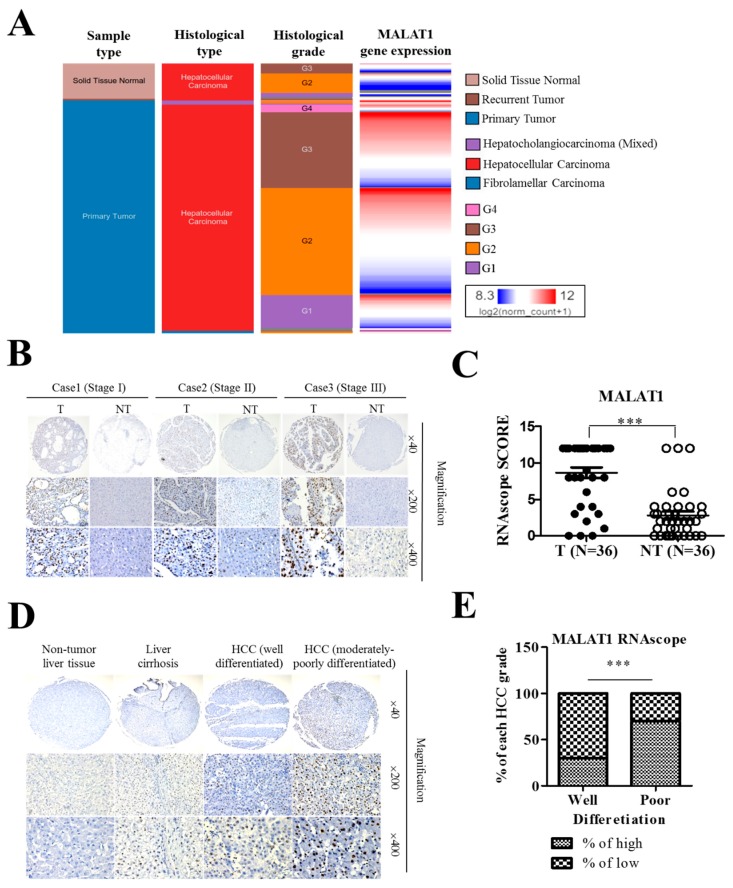
LncRNA MALAT1 overexpression in liver cancer positively correlates with poor cellular differentiation status and disease progression. (**A**) In the TCGA liver cancer (LIHC) cohort (n = 438) significantly positive correlation between MALAT1 expression and cellular differentiation (histologic) grade was observed. G1, grade 1 = well differentiated; G2, grade 2 = moderately differentiated; G3, grade 3 = poorly differentiated; G4, grade 4 = undifferentiated/anaplastic. (**B**) Representative RNAscope images (T = Tumor; NT = Non-tumor) and (**C**) Boxplots graph depicting the manual scores showing positive correlation between increased expression of MALAT1 and HCC pathological stage. (**D**) Representative RNAscope images and (**E**) stacked histogram showing the differential expression of MALAT1 in non-tumor liver tissue, liver cirrhosis, well-differentiated HCC and moderately-poorly differentiated HCC tissues. *** *p* < 0.001.

**Figure 3 cells-09-01020-f003:**
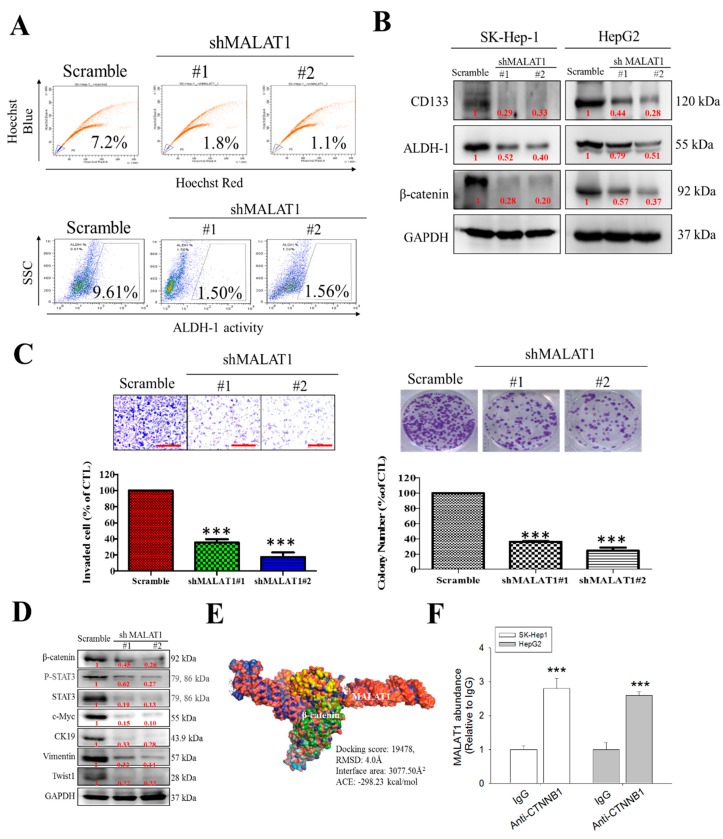
MALAT1 expression modulates HCC oncogenicity and stemness via interaction with β-catenin. (**A**) Flow cytometric analysis showed that MALAT1 down-regulation resulted in decreased percentage of side-population cells (upper panel) and ALDH1+ cells (lower panel) in primary liver cancer cells. (**B**) The inhibitory effect of silencing MALAT1 on the expression of CD133, ALDH1 and β-catenin proteins in SK-Hep1 and HepG2 cells demonstrated by western blot (**C**) MALAT1-silenced SK-Hep1 cells were observed to exhibit a marked decrease in their invasiveness (left) and colony-forming ability (right). (**D**) Representative western blots data showing the inhibitory effect of MALAT1 silencing on the expression level of β-catenin, p-STAT3, STAT3, c-Myc, CK19, vimentin and Twist1 proteins in SK-Hep1 cells. GAPDH served as loading control. (**E**) Molecular visualization of the direct interaction between MALAT1 and β-catenin using PyMOL academic edition. (**F**) Graph showing enhanced MALAT1 abundance upon immunoprecipitation with β-catenin in SK-Hep1 and HepG2 cells. *** *p* < 0.001.

**Figure 4 cells-09-01020-f004:**
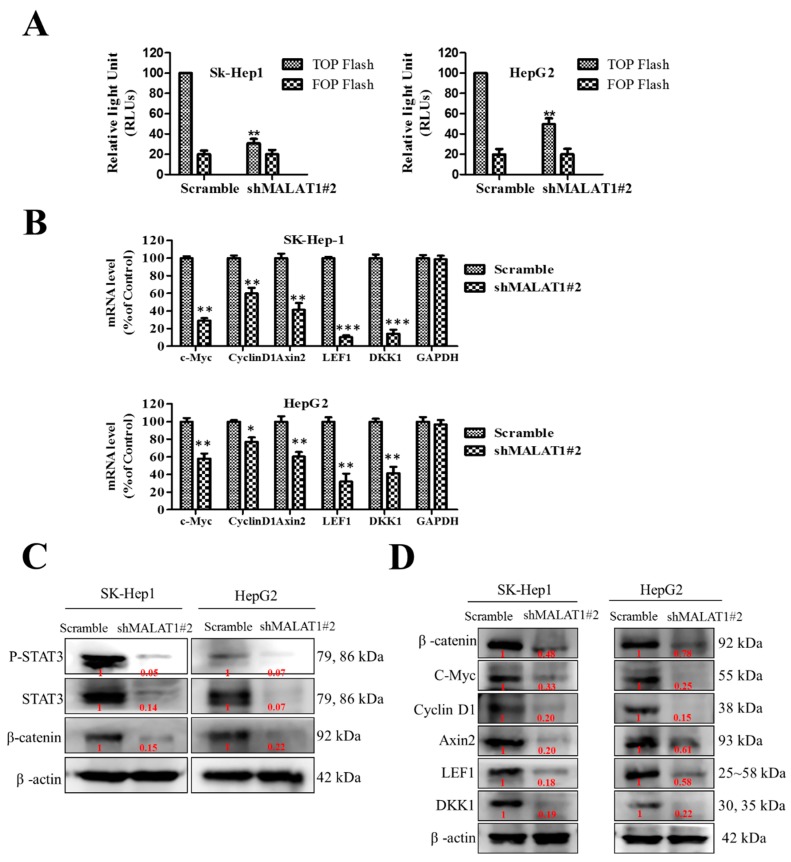
MALAT1 oncogenic activity in HCC is mediated by the Wnt/β-catenin signaling pathway. (**A**) MALAT1 down-regulation led to a significantly decreased transcriptional activity of β-catenin in both SK-Hep1 and HepG2 cells, as assessed by the TOPFlash/FOPFlash luciferase assay (**B**) Graph showing the suppression of c-Myc, cyclin D1, Axin2, LEF1 and DKK1 mRNA levels in MALAT1-silenced SK-Hep1 (upper panel) or HepG2 (lower panel), compared to control. Representative western blot images of the effect of shMALAT1#2 on the expression of (**C**) STAT3 and β-catenin or (**D**) β-catenin, c-Myc, cyclin D1, Axin2, LEF1 and DKK1 proteins in SK-Hep1 or HepG2 cells. β-actin served as loading control. ** *p* < 0.01, *** *p* < 0.001.

**Figure 5 cells-09-01020-f005:**
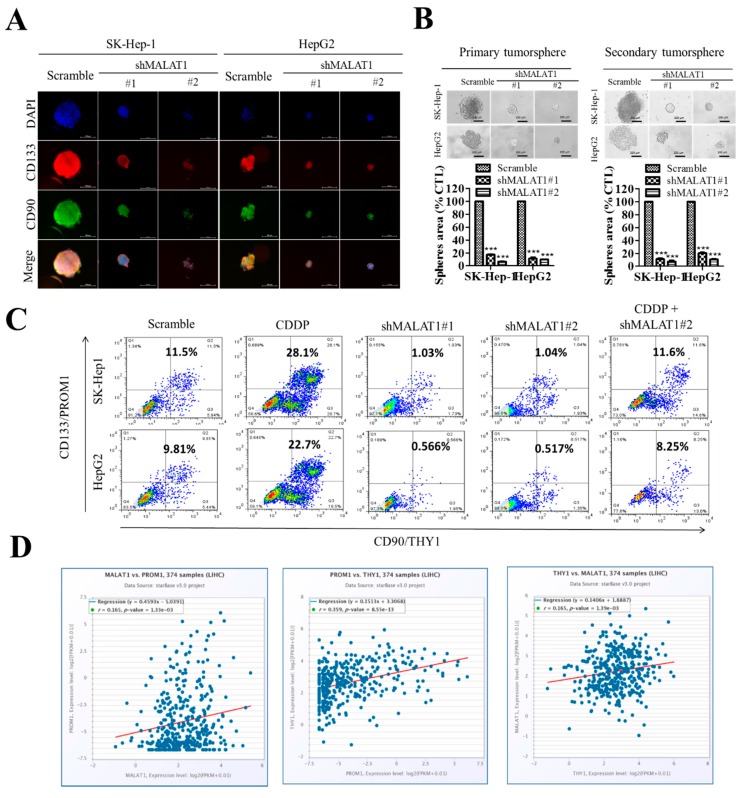
Silencing MALAT1 is associated with reduced CD133^high^CD90^high^ HCC population with suppressed HCC tumorsphere formation in vitro. (**A**) Photo-images showing the effect of shMALAT1#1 and shMALAT1#2 on the size of tumorspheres formed and on the expression of CD133 and CD90 (left panel). Histograms show the effect of shMALAT1 #1 and #2 on number of tumorspheres formed (right panel). (**B**) SK-Hep1 or HepG2 cells transfected with shMALAT1#1 and shMALAT1#2 exhibited decreased HCC tumorsphere size and number in both primary and secondary generation tumorsphere. (**C**) Comparative flow-cytometry analysis CD133 and CD90 positivity in SK-Hep1 or HepG2 cells exposed to CDDP or transfected with shMALAT1#1 and shMALAT1#2. (**D**) Graphical representation of the correlation between MALAT1, CD133, and CD90. CDDP, cisplatin; *** *p* < 0.001.

**Figure 6 cells-09-01020-f006:**
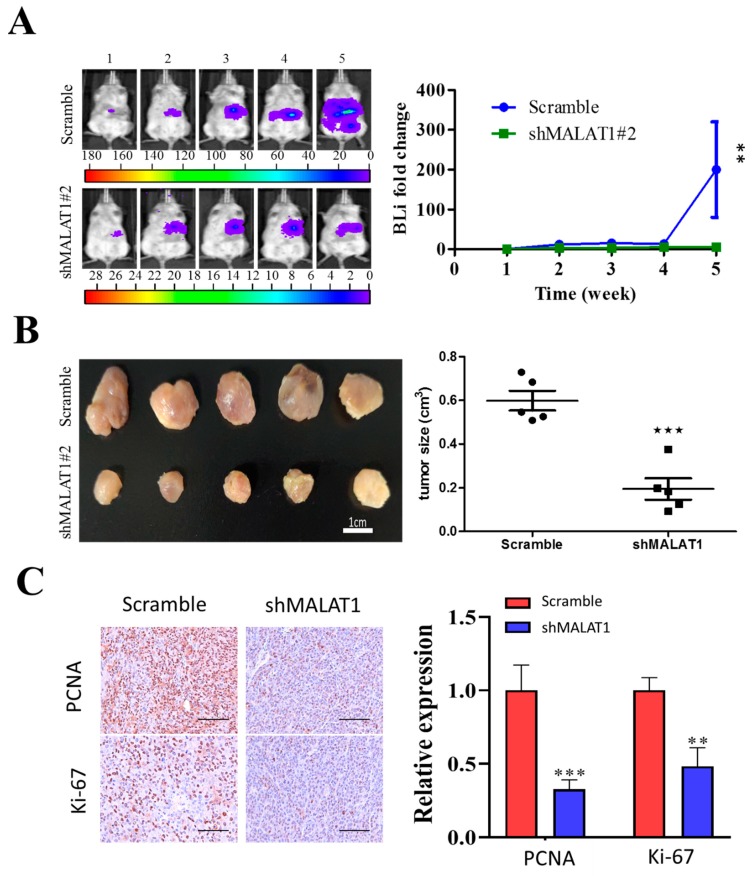
MALAT1 depletion impedes xenograft growth of HCC cells. Luciferase-expressing SK-Hep1 cells with or without MALAT1 depletion were injected into NOD/SCID mice. (**A**) Representative images comparing the tumorigenic potential in NOD/SCID mice bearing scrambled control or shMALAT1#2 SK-Hep1 tumors over a period of 5 weeks using bioluminescence imaging technique. Semi-quantitative analysis of the change in bioluminescent intensity (fold change) over time curve showed a significantly lower tumor burden was in the shMALAT1 group. (**B**) Photographs of the tumor samples harvested, and tumor volumes measured at week 5, demonstrating the shMALAT1 tumors were significantly smaller than the control counterparts. (**C**) Representative histochemical staining of Ki-67 and PCNA in primary HCC tumor cells from control and MALAT1 shRNA-transfected mice. The percentage of tumor cells that were positive for Ki-67 and PCNA was calculated by counting 10 visual fields at high magnification. Scale bars, 50 μm. Bioluminescence intensity, BLI; PCNA, proliferating cell nuclear antigen. Data are mean ± SEM **, *p* < 0.01; ***, *p* < 0.001.

**Figure 7 cells-09-01020-f007:**
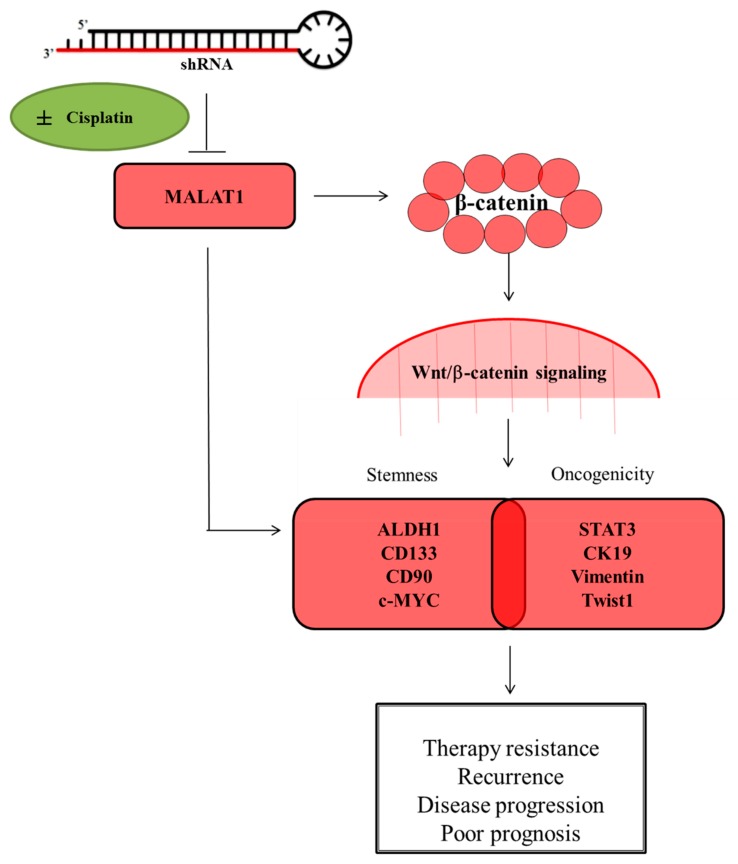
Schematic abstract of the oncogenic and CSCs-promoting role of MALAT1 in HCC. The molecular targeting of the long non-coding RNA MALAT1 suppresses the stemness and metastatic phenotypes of hepatocellular carcinoma cells by modulating the Wnt/β-catenin signaling pathway.

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
