# Peer review of "Targeting the Epigenetic Non-Coding RNA MALAT1/Wnt Signaling Axis as a Therapeutic Approach to Suppress Stemness and Metastasis in Hepatocellular Carcinoma"

_cells, 2020, doi:10.3390/cells9041020_

Round 1
Reviewer 1 Report
The findings are quite important for lncRNA-cancer field. MALAT1 is of significant interest in many cancers and this study shows how it affects HCC. However, some revisions are required to enhance the quality of the manuscript.
Comment 1: In the methods section, page 3, in the “MALAT1 silencing” paragraph, the authors should mention how they designed the shRNA sequences.
Comment 2: Page 3, line 129, in the “Western Blot” section, dilutions of all primary antibodies used should be noted. Also mention how the blots were quantified. Graph is not plotted in the result section with the quantified values.
Comment 3: Page 3, line 137 beta-actin has a typo.
Comment 4: Page 4, line 178, the cell number will be 3.5x104 , but the 4 is not in superscript. This issue is found many times in the manuscript, for example, page 5, lines 193 and 210; page 6, line 234.
Comment 5: Page 5, in the “Colony forming’ section, it appears that the number of cells plated was too high and also incubated for an extended period of time. Generally single cells will form colonies in 7 – 14 days when they are plated starting from very high dilutions of 100-200 cells per well. This varies from cell to cell. Did the authors perform increased dilutions to optimize the number of cells to be plated? There is however no doubt that the colony forming potential of cells is reduced on knocking down MALAT1. But quantification of colonies becomes difficult if cells proliferates rapidly..
Comment 6: In the methods, for invasion, transwell, colony formation and tumor formation assays, the authors have mentioned wild type and shMALAT1 cells, but they have shown scrambled controls in the results. The authors should mention in “methods” section also that cells transfected with scrambled shRNA were used as control for all these experiments and not only wild type cells.
Comment 7: The authors have not shown knock down percentage of MALAT1 after transfection with the shRNAs.
Comment 8: Page 6, line 244 the word “experimental” has been misspelled.
Comment 9: Page 15, Figure 5C, more detailed explanation of how the correlations were generated should be provided in the legends or in the methods section. In the in vivo data, Figure 5D, are these five individual mice in two groups or does the image represents one mouse from each group that was tested for tumor growth over a period of 5 weeks? If the former is correct, there appears to be no significant difference in tumor growth, however, if the latter is correct, images of all 5 mice from each group should be provided. Author should also provide the survival data of mice and also tumor size. IHC to show decreased proliferation in shMALAT1 tumor tissue from mice should be shown by PCNA staining and should be tested for expression of MALAT1.
Author Response
Response to Reviewers:
Point-by-point responses to reviewer’s comments - Reviewer 1:
We thank the reviewer for carefully reading our manuscript and providing valuable comments. We believe making use of all these comments has further helped improve the quality and appeal of our work, as well as strengthened the manuscript. Below are our point-by-point responses.
Q1.1. The findings are quite important for lncRNA-cancer field. MALAT1 is of significant interest in many cancers and this study shows how it affects HCC. However, some revisions are required to enhance the quality of the manuscript.
A1.1. We thank the reviewer for taking time to read our manuscript and for the critiques and suggestions made in order to help us improve the quality of our work. In this revised manuscript, we have made effort to make address all the comments and suggestions.
Q1.2. In the methods section, page 3, in the “MALAT1 silencing” paragraph, the authors should mention how they designed the shRNA sequences.
A1.2. We sincerely thank the reviewer for these comments. We have now revised our manuscript to address these issues raised by the reviewer. Please kindly see our revised Materials & Methods section, Lines 115-124.
2.3. MALAT1 silencing
The shRNA specifically targeting MALAT1 was constructed by CRISPR Gene Targeting Core Lab (Taipei Medical University, Taiwan). For MALAT1 silencing, SK-Hep1 and HepG2 cells were transfected with MALAT1 shRNA (shRNA#1: Forward 5’-CCGGAAAGCCCTGA ACTATCACACTCTCGAGAGTGTGATAGTTCAGGGCTTTTTTG-3’; Reverse 5’-AATTCAAAAAAAAGCCCTGAA CTATCACACTCTCGAGA GTGT GATAGTTCAGGGCTTT -3’ or shRNA#2: Forward 5’- CCGGAATCTGTAAGCAGTTTGTATGCTCGAGCATACAAA CTGCTTACAGATTTTTTTG-3’; Reverse 5’- AATTCAAAAAAA TCTGTAAGCAGTTTGTATGCT CGAGCATACAAACTGCTTACAGATT -3’) or vector, then the stably transfected shMALAT SK-Hep1 and HepG2 cells were selected with 1μg/ml puromycin.
Q1.3. Page 3, line 129, in the “Western Blot” section, dilutions of all primary antibodies used should be noted. Also mention how the blots were quantified. Graph is not plotted in the result section with the quantified values.
A1.3. We thank the reviewer for this observation. As suggested, we have now included the dilutions of all primary antibodies in our revised manuscript. Please kindly see our revised Materials & Methods section, Lines 125-145.
2.4. Western blot analysis
After washing cells with PBS twice and lysing with ice-cold RIPA lysis buffer (#20-188, Sigma-Aldrich Co., St. Louis, MO, USA), the total protein lysate from the wild-type or shMALAT1 HCC cells was centrifuged and pellet collected. Protein concentration was then quantified using Pierce™ BCA protein assay kit (#23227, Thermo Fisher Scientific Inc., Waltham, MA, USA). Equal amount of protein lysate from each sample was run on 10% sodium dodecyl sulfate polyacrylamide gel electrophoresis (SDS-PAGE) then protein transferred to Polyvinylidene difluoride (PVDF) membranes which were blocked in 1X PBS containing 5% skimmed milk, and then incubated with the specific primary antibodies against β-catenin (1:1000, β-Catenin (6B3) Rabbit mAb #9582P, Sigma), CD133 (1:1000, #MAB4310, Sigma), ALDH-1 (1:1000, (D4R9V) Rabbit mAb #12035, Sigma), c-Myc (1:1000, c-Myc Antibody (9E10) (sc-40), Sigma), CK19 (1:1000, monoclonal antibody SAB3300019, Sigma), Stat3 (1:1000, Stat3 (79D7) Rabbit mAb #4904, Sigma), vimentin (1:1000, Anti-Vimentin antibody (ab137321), Sigma), Twist1 (1:1000, Twist (Twist2C1a) Antibody sc-81417, Sigma), cyclin D1 (1:1000, Cyclin D1 Antibody (A-12) (sc-8396), Sigma), Axin2 (1:1000, Axin2 Antibody SAB3500619, Sigma), LEF1 (1:1000, LEF1 (C12A5) Rabbit mAb #2230, Sigma), or DKK1 (1:1000, DKK1 Antibody #4687, Sigma) at 4 °C overnight for protein detection in Supplementary Table S1. After overnight probing, protein bands were identified using anti-mouse or anti-rabbit horseradish peroxidase (HRP)-linked secondary antibody at room temperature for 1 h. The signal was detected using UVP® imaging system (Analytik Jena US LLC., Upland, CA, USA). β-actin (1:10000, 8H10D10, Mouse mAb #3700) was used as loading control. The gray value was quantified and analyzed using Image J software. The experiment was repeated 3 times.
Q1.4. Page 3, line 137 beta-actin has a typo.
A1.4. We thank the reviewer for this keen observation. The typo errors are now corrected. Once again, we thank the reviewer.
Q1.5. Page 4, line 178, the cell number will be 3.5x104, but the 4 is not in superscript. This issue is found many times in the manuscript, for example, page 5, lines 193 and 210; page 6, line 234.
A1.5. We thank the reviewer for this keen observation. The cell number superscript is now corrected. Once again, we thank the reviewer.
Q1.6. Page 5, in the “Colony forming’ section, it appears that the number of cells plated was too high and also incubated for an extended period of time. Generally single cells will form colonies in 7 – 14 days when they are plated starting from very high dilutions of 100-200 cells per well. This varies from cell to cell. Did the authors perform increased dilutions to optimize the number of cells to be plated? There is however no doubt that the colony forming potential of cells is reduced on knocking down MALAT1. But quantification of colonies becomes difficult if cells proliferates rapidly.
A1.6. We thank the reviewer for this observation. As suggested, we have now included the in newly Figure 3C in our revised manuscript. Please kindly see our revised Materials & Methods section, Lines 185-192.
2.8. Colony formation assay
Approximately 2x102 scrambled shRNA or shMALAT1 SK-Hep1 and HepG2 cells seeded into a 6-well cell culture plate were incubated for 10 – 12 days at 37 °C in humidified 5% CO2 atmosphere. Cells were then washed twice with PBS, fixed in cold methanol, stained with 0.005 % crystal violet, washed several times and air-dried. The total number of colonies formed with diameter ≥ 100 μm in each well was estimated over 5 randomly selected visual fields in assays performed 4 times in triplicates. The visible colonies were manually counted, and the rate of colony formation was calculated with the following equation: (number of colonies/numbers of seeded cells) × 100%.
Q1.7. In the methods, for invasion, transwell, colony formation and tumor formation assays, the authors have mentioned wild type and shMALAT1 cells, but they have shown scrambled controls in the results. The authors should mention in “methods” section also that cells transfected with scrambled shRNA were used as control for all these experiments and not only wild type cells.
A1.7. We thank the reviewer for this keen observation. The scrambled shRNA is now corrected in revised in our revised manuscript. Please kindly see our revised Materials & Methods section, Lines 176-201.
2.7. Matrigel invasion assay
For analysis of the invasion potential of scrambled shRNA or shMALAT1 HCC cells, the 24-well plate Transwell system was used. 3.5 × 104 cells were seeded onto the upper chambers of the inserts (BD Bioscience, 8 μm pore size) containing serum-free media, while the lower chambers contained media with 10 % FBS serving as chemo-attractant. Medium was discarded after 24 h incubation, and then non-invaded cells remaining on the upper side of the inserts were removed with sterile cotton swabs while invaded cells underneath the filter membranes were fixed with 3.7 % formaldehyde for 1 h and stained with crystal violet. The invaded cells were visualized and evaluated under microscope.
2.8. Colony formation assay
Approximately 2x102 scrambled shRNA or shMALAT1 SK-Hep1 and HepG2 cells seeded into a 6-well cell culture plate were incubated for 10 – 12 days at 37 °C in humidified 5% CO2 atmosphere. Cells were then washed twice with PBS, fixed in cold methanol, stained with 0.005 % crystal violet, washed several times and air-dried. The total number of colonies formed with diameter ≥ 100 μm in each well was estimated over 5 randomly selected visual fields in assays performed 4 times in triplicates. The visible colonies were manually counted, and the rate of colony formation was calculated with the following equation: (number of colonies/numbers of seeded cells) × 100%.
2.9. Tumorsphere formation assays
HCC tumorspheres were generated from scrambled shRNA or shMALAT1 SK-Hep1 and HepG2 single-cell suspension. SK-Hep1 or HepG2 cells were seeded at a density of 5 × 104 per well into Corning® Costar® ultra-low attachment 6-well plates (Corning, NY, USA) containing growth factors for CSCs enrichment stem cell medium (Nutristem-XF, Biological Industries, Israel) and incubated at 37˚C in humidified 5% CO2 incubator for 5 - 7 days, followed by visualization of tumorspheres (diameter ≥ 100 μm) under an inverted microscope. Tumorsphere size, quantity and formation efficiency were evaluated. Tumorsphere formation efficiency (TFE) was evaluated using the formula: TFE = (number of sphere formed/number of single cells plated) x 100.
Q1.8. The authors have not shown knock down percentage of MALAT1 after transfection with the shRNAs.
A1.8. We thank the reviewer for this keen observation. The knock down percentage of MALAT1 is now corrected in revised in our revised manuscript. Please kindly see our revised Materials & Methods section, Lines 115-124.
2.3. MALAT1 silencing
The shRNA specifically targeting MALAT1 was constructed by CRISPR Gene Targeting Core Lab (Taipei Medical University, Taiwan). For MALAT1 silencing, SK-Hep1 and HepG2 cells were transfected with MALAT1 shRNA (shRNA#1: Forward 5’-CCGGAAAGCCCTGA ACTATCACACTCTCGAGAGTGTGATAGTTCAGGGCTTTTTTG-3’; Reverse 5’-AATTCAAAAAAAAGCCCTGAACTATCACACTCTCGAGAGTGT GATAGTTCAGGGCTTT -3’ or shRNA#2: Forward 5’- CCGGAATCTGTAAGCAGTTT GTATGCTCGAGCATACAAA CTGCTTACAGATTTTTTTG-3’; Reverse 5’- AATTCAAAAAAA TCTGTAAGCAGTTTGTATGCT CGAGCATACAAACTGCTTACAGATT -3’) or vector, then the stably transfected shMALAT SK-Hep1 and HepG2 cells were selected with 1μg/ml puromycin.
Q1.9. Page 6, line 244 the word “experimental” has been misspelled.
A1.9. We thank the reviewer for this keen observation. The typo errors are now corrected. Once again, we thank the reviewer.
Q1.10. Page 15, Figure 5C, more detailed explanation of how the correlations were generated should be provided in the legends or in the methods section. In the in vivo data, Figure 5D, are these five individual mice in two groups or does the image represents one mouse from each group that was tested for tumor growth over a period of 5 weeks? If the former is correct, there appears to be no significant difference in tumor growth, however, if the latter is correct, images of all 5 mice from each group should be provided. Author should also provide the survival data of mice and also tumor size. IHC to show decreased proliferation in shMALAT1 tumor tissue from mice should be shown by PCNA staining and should be tested for expression of MALAT1.
A1.10. We thank the reviewer for this observation. As suggested, we have now included the Photographs of the tumor samples and IHC assay in newly Figure 6 in our revised manuscript. Please kindly see our revised results section, Lines 452-464.
3.6. MALAT1 is required for xenograft tumorigenesis and tumor growth of HCC cells in vivo
Having established that MALAT1 is required for the growth of HCC cells in vitro, we decided to determine if MALAT1 was also required for in vivo growth of HCC cells. According to in vivo mice tumor xenograft models, we observed significant lower bioluminescent intensity in the mice bearing shMALAT1#2 tumors as compared to the control. For instance, the fold change in bioluminescence in the shMALAT1#2 group was approximately 198.5-fold (p < 0.01) lower than that in the control counterpart at week 5 (Figure 6A). As shown in Figure 6A, MALAT1‐depleted xenografts grew much slower than controls. At the end of experiment (week 5), control tumors had grown to much larger sizes than that of MALAT1‐depleted ones (Figure 6B). Moreover, MALAT1‐depleted xenografts tumors displayed decreased Ki67 and PCNA expression, indicating these tumors were more proliferative inhibition than control tumors (Figure 6C). These data indicate a role for silencing MALAT1 in the suppression of HCC-SC-induced and/or maintained tumorigenicity and cancer stemness.
Please kindly see our revised Figure 6 legend, Lines 481-493.
Figure 6. MALAT1 depletion impedes xenograft growth of HCC cells. Luciferase‐expressing SK-Hep1 cells with or without MALAT1 depletion were injected into NOD/SCID mice. (A) Representative images comparing the tumorigenic potential in NOD/SCID mice bearing scrambled control or shMALAT1#2 SK-Hep1 tumors over a period of 5 weeks using bioluminescence imaging technique. Semi-quantitative analysis of the change in bioluminescent intensity (fold change) over time curve showed a significantly lower tumor burden was in the shMALAT1 group. (B) Photographs of the tumor samples harvested, and tumor volumes measured at week 5, demonstrating the shMALAT1 tumors were significantly smaller than the control counterparts. (C) Representative histochemical staining of Ki-67 and PCNA in primary HCC tumor cells from control and MALAT1 shRNA-transfected mice. The percentage of tumor cells that were positive for Ki-67 and PCNA was calculated by counting 10 visual fields at high magnification. Scale bars, 50 μm. Bioluminescence intensity, BLI; PCNA, proliferating cell nuclear antigen. Data are mean ± SEM **, P < 0.01; ***, P < 0.001.
Please kindly see our revised Material and Methods section, Lines 247-252.
2.15. Mice tumor xenograft studies
All tumor xenograft studies were approved by the institutional research ethics committee and the Institutional Animal Care and Use Committee (Approval number: LAC-2018-0572). 4–6-week-old female NOD/SCID mice were purchased from the BioLASCO (BioLASCO Taiwan Co., Ltd. Taipei, Taiwan). Mice were inoculated with 1 × 106 shMALAT1#2 SK-Hep1 tumorsphere-derived cells (N=5), while the control mice were inoculated with the scrambled control SK-Hep1 tumorsphere-derived cells (N=5). Tumor growth was then monitored by in vivo bioluminescence imaging (IVIS200 imaging system, Caliper Life Sciences Inc., Hopkinton, MA, USA) for 5 weeks. Post-experiment, all animals were humanely sacrificed by tumor dislocation and the tumor samples were harvested for further analyses.

Reviewer 2 Report
General comment
Chang et al. aimed to study the role of MALAT1 in hepatocellular carcinoma (HCC). They used publicly available datasets from gene expression analysis derived from patient liver tissues (TCGA liver cancer cohort, Oncomine Wurmbach liver dataset) and human hepatoma cells lines to assess the expression of defined genes, including beta-catenin, Stat3, CD133 and ALDH1 following MALAT1 silencing. The role of MALAT1 silencing for hepatoma cell tumorsphere formation and invasiveness in vitro as well as tumor growth in immunodeficient mice was also studied.
HCC is a major cause of cancer death to date and novel therapeutic strategies are urgently needed. The study of the role of lncRNA in HCC pathogenesis and their therapeutic potential is of great interest. MALAT1 has been previously linked to HCC and deserves further characterization.
The present manuscript is difficult to read. The data are not well described and interpreted. Several conclusions appear to be overstated and not supported by the shown data.
Specific comments in the order of appearance in the manuscript:
- Lines 90 and 108 (as well as other parts of the manuscript where HepG2 celsl are refered to as HCC cells): "human hepatoma cell lines" (not HCC cell lines as HepG2 cells are hepatoblastoma cells)
- Line 140: what is the "TMU-SHH HCC cohort". The reviewer could not find a description of the patients in the manuscript nor a reference.
- Line 203: the RIP experiment needs some explanations (which cells were used? How was RNA isolated? how much RNA was used for the assay?).
- Figure 1B: why were Huh7 cells chosen as "reference"? It would make more sense to compare different hepatoma cell lines to THLE-2 cells.
- Figure 1: legend indicates statistical analysis/significance but no symbols are displayed in the panels.
- It is impossible to understand some of the Figures by only reading the legend and sometimes even with the data description in the results section. One example is Figure 4A. The authors should explain why/how they performed the assay and guide the reader.
- Figure 2C: is not a "dot-and-wisker plot" (used for regression analysis). The legend should describe what the horizontal bar describes (mean? median?) and whether standard deviation (or else?) is represented.
- Line 317: "vital role" is an overstatement.
- Section 3.3: this paragraph needs more details and a better data analysis. Line 337: the MALAT1 sh clones need to be introduced/characterized. The reviewer could not find a statement how these cells were chosen, how much MALAT1 they express as compared to the parental cells, etc... Line 343: "silencing MALAT1 reduced ALDH1 activity by 8,11%" should be interpreted in a different way (rephrased as before for Hoechst data). The experiments shown in Figure 3C need to be better described in the text. The rationale for analyzing the genes assessed in Figure 3D need to be clearly stated.
- Figure 3A upper panel/lines 339: the figures in the panels and the text are different; this is confusing. The size of the panels could be increased. The labeling appears to be blurred.
- Figure 3C: the statistical analysis appears to be incorrectly indicated unless the authors compared both MALAT1 sh clones? The size of the upper panels could be increased.
- Figure 3F: the authors should more carefully interpret their data. The sentence in lines 370-371 is an overstatement. RIP does not demonstrate a direct interaction between MALAT1 and beta-catenin. This assay demonstrates that both molecules can be coimmunoprecipitated but other interacting molecules could be involved.
- Section 3.4: the experiments need to be better described. The conclusion appears to be overstated (not all protein levels appear to be significantly reduced).
- Figure 5B, section 3.5: this needs to be better described. The conclusion does not appear to be in line with the shown data. The authors are mixing data from CDDP treatment and MALAT1 silencing to conclude that "Interestingly, we observed that contrary to conventional knowledge, CDDP increased the population of CD133highCD90high SK-Hep1 429 (28.1%) or HepG2 (22.7%) cells, compared to vehicle-treated scrambled control (Figure 5B, upper), suggesting a CDDP-induced HCC-SCs phenotype. However, this enhanced CD133/90 positivity was markedly repressed in the shMALAT1#1 and shMALAT1#2 HepG2 or SK-Hep1 cells (Figure 5B, lower)." Given the experimental set up the authors cannot link CDDP and MALAT1 silencing data as they did not add CDDP to MALAT1 sh cells. Thus the conclusion (line 533) does not appear to be supported by the shown data.
- To strengthen their conclusions functional rescue experiments should be included.
- Figure 5C: "significant positive correlation" is an overstatement.
- General comment on figures to increase their readability: FACS panels should be improved (increase size, provide clear labeling). The molecular weight of the proteins detected by Western blot should be shown.
Author Response
Point-by-point responses to reviewer’s comments - Reviewer 2:
We would like to thank the reviewer for the thorough reading of our manuscript as well as their valuable comments. We believe all comments are borne out of good faith, and thus, have tried to address their comments conscientiously and feel that they have further improved the readability and appeal of our work, as well as strengthened the manuscript. Below are our point-by-point responses.
Q2.1. Chang et al. aimed to study the role of MALAT1 in hepatocellular carcinoma (HCC). They used publicly available datasets from gene expression analysis derived from patient liver tissues (TCGA liver cancer cohort, Oncomine Wurmbach liver dataset) and human hepatoma cells lines to assess the expression of defined genes, including beta-catenin, Stat3, CD133 and ALDH1 following MALAT1 silencing. The role of MALAT1 silencing for hepatoma cell tumorsphere formation and invasiveness in vitro as well as tumor growth in immunodeficient mice was also studied. HCC is a major cause of cancer death to date and novel therapeutic strategies are urgently needed. The study of the role of lncRNA in HCC pathogenesis and their therapeutic potential is of great interest. MALAT1 has been previously linked to HCC and deserves further characterization. The present manuscript is difficult to read. The data are not well described and interpreted. Several conclusions appear to be overstated and not supported by the shown data.
A2.1. We thank the reviewer for taking time to read our manuscript and for the critiques and suggestions made in order to help us improve the quality of our work. In this revised manuscript, we have made effort to make address all the comments and suggestions.
Q2.2. Lines 90 and 108 (as well as other parts of the manuscript where HepG2 cells are refered to as HCC cells): "human hepatoma cell lines" (not HCC cell lines as HepG2 cells are hepatoblastoma cells)
A2.2. We thank the reviewer for the keen observation. As suggested by the reviewer, we have now corrected the HepG2 cells are hepatoblastoma cells in our revised manuscript. Please kindly see our revised Materials and Methods section, Lines 107-114.
2.2. Cell culture
The human HCC cell lines, Huh7, Mahlavu, SK-Hep1 and hepatoblastoma cells HepG2, as well as non-tumor liver cell line THLE-2 were purchased from ATCC and cultured in DMEM or RPMI1640 supplemented with 10% (v/v) heat-inactivated fetal bovine serum (FBS), Penicillin (100 IU/mL) and Streptomycin (100 μg/mL), in humidified 5% CO2 atmosphere at 37 °C. Cells were sub-cultured when they attained a confluence ≥ 90 and media changed every 48-72 h. Establishment and growth of primary HCC culture cells were performed strictly according to protocol by Cheung PF et al. [12].
Q2.3. Lines 90 and 108 (as well as other parts of the manuscript where HepG2 cells are refered to as HCC cells): "human hepatoma cell lines" (not HCC cell lines as HepG2 cells are hepatoblastoma cells)
A2.3. We thank the reviewer for the keen observation. As suggested by the reviewer, we have now corrected the HepG2 cells are hepatoblastoma cells in our revised manuscript. Please kindly see our revised Materials and Methods section, Lines 107-114.
2.2. Cell culture
The human HCC cell lines, Huh7, Mahlavu, SK-Hep1 and hepatoblastoma cells HepG2, as well as non-tumor liver cell line THLE-2 were purchased from ATCC and cultured in DMEM or RPMI1640 supplemented with 10% (v/v) heat-inactivated fetal bovine serum (FBS), Penicillin (100 IU/mL) and Streptomycin (100 μg/mL), in humidified 5% CO2 atmosphere at 37 °C. Cells were sub-cultured when they attained a confluence ≥ 90 and media changed every 48-72 h. Establishment and growth of primary HCC culture cells were performed strictly according to protocol by Cheung PF et al. [12].
Q2.4. Line 140: what is the "TMU-SHH HCC cohort". The reviewer could not find a description of the patients in the manuscript nor a reference.
A2.4. We thank the reviewer for the keen observation. As suggested by the reviewer, we have now deleted the TMU-SHH HCC cohort in our revised manuscript. Please kindly see our revised Materials and Methods section, Lines 146-157.
2.5. Immunohistochemistry (IHC) analysis
All human tissues were obtained from surgical resection specimens of HCC patients at Taipei Medical University-Shuang-Ho Hospital (New Taipei City, Taiwan). Tissue microarray (TMA) slides were established, then heat-based antigen retrieval was performed in EDTA-containing buffer, sections blocked with 5% bovine serum albumin (BSA)/1% HISS/0.1% Tween20 solution, and incubated with primary recombinant antibody against MALAT1 (1:400 dilution; #MOB-4044z, Creative Biolabs, NY, USA) overnight, at 4˚C. MALAT1 immunoreactivity/positivity was detected using the mouse IgGk light chain binding protein conjugated to horseradish peroxidase m-IgGβ BP-HRP (#sc-516102; Santa Cruz Biotechnology, Inc., Santa Cruz, CA, USA) and the EXPOSE mouse and rabbit specific HRP/DAB detection IHC kit (#ab80436, Abcam plc., Cambridge, MA, USA). This study was approved by the Institutional Human Research Ethics Review Board (TMU-JIRB No. 201302016) of Taipei Medical University.
Q2.5. Line 203: the RIP experiment needs some explanations (which cells were used? How was RNA isolated? how much RNA was used for the assay?).
A2.5. We thank the reviewer for the keen observation. As suggested by the reviewer, we have now corrected the RIP experiment protocol in our revised manuscript. Please kindly see our revised Materials and Methods section, Lines 205-214.
2.11. RNA immunoprecipitation
LncBase Predicted v.2 bioinformatics tools (http://carolina.imis.athena-innovation.gr/) was used to predict the potential interaction of MALAT1 and CTNNB1. RNA immunoprecipitation (RIP) analysis was performed using EZ-Magna RIP™ RNA-Binding Protein Immunoprecipitation Kit (#17-701; Sigma-Aldrich Corporation, St. Louis, MO, USA) strictly following the manufacturer’s recommended protocol. The RNA from the whole cell lysates and the RIP fractions were extracted with TRIzol according to the manufacturer's instructions. The relative mRNA expression levels of MALAT1 and CTNNB1 were determined using RT-qPCR analysis, as described above. The relative mRNA enrichment in the RIP fractions was computed based on the ratio of relative mRNA levels in the RIP fractions and the relative mRNA levels in the whole cell lysates (input).
Q2.6. Figure 1: legend indicates statistical analysis/significance, but no symbols are displayed in the panels.
A2.6. We thank the reviewer for this observation. As suggested, we have now included the statistical analysis in newly Figure 1C in our revised manuscript. Please kindly see the newly Figure 1C.
Q2.7. Figure 1: legend indicates statistical analysis/significance, but no symbols are displayed in the panels.
A2.7. We thank the reviewer for this observation. As suggested, we have now included the statistical analysis in newly Figure 1C in our revised manuscript. Please kindly see the newly Figure 1C.
Q2.8. It is impossible to understand some of the Figures by only reading the legend and sometimes even with the data description in the results section. One example is Figure 4A. The authors should explain why/how they performed the assay and guide the reader.
A2.8. We thank the reviewer for this observation. As suggested, we have now included the statistical analysis in newly Figure 4A in our revised manuscript. Please kindly see the newly Figure 4A.
Q2.9. Figure 2C: is not a "dot-and-wisker plot" (used for regression analysis). The legend should describe what the horizontal bar describes (mean? median?) and whether standard deviation (or else?) is represented.
A2.9. We thank the reviewer for this observation. As suggested, we have now included the statistical analysis in newly Figure 2C in our revised manuscript. Please kindly see the newly Figure 2C legend.
Figure 2. LncRNA MALAT1 overexpression in liver cancer positively correlates with poor cellular differentiation status and disease progression. (A) In the TCGA liver cancer (LIHC) cohort (n = 438) significantly positive correlation between MALAT1 expression and cellular differentiation (histologic) grade was observed. G1, grade 1 = well differentiated; G2, grade 2 = moderately differentiated; G3, grade 3 = poorly differentiated; G4, grade 4 = undifferentiated/anaplastic. (B) Representative RNAscope images (T=Tumor; NT=Non-tumor) and (C) Boxplots graph depicting the manual scores showing positive correlation between increased expression of MALAT1 and HCC pathological stage. (D) Representative RNAscope images and (E) stacked histogram showing the differential expression of MALAT1 in non-tumor liver tissue, liver cirrhosis, well-differentiated HCC and moderately-poorly differentiated HCC tissues. *p < 0.05, **p< 0.01, ***p< 0.001.
Q2.9. Line 317: "vital role" is an overstatement.
A2.9. We thank the reviewer for this observation. As suggested, we have now corrected the sentances in our revised manuscript. Please kindly see the line 324-326.
These data are indicative of the aberrant MALAT1 expression in HCC progression and suggestive of its probable role as a biomarker of poor cellular differentiation in HCC.
Q2.10. Section 3.3: this paragraph needs more details and a better data analysis. Line 337: the MALAT1 sh clones need to be introduced/characterized. The reviewer could not find a statement how these cells were chosen, how much MALAT1 they express as compared to the parental cells, etc... Line 343: "silencing MALAT1 reduced ALDH1 activity by 8,11%" should be interpreted in a different way (rephrased as before for Hoechst data). The experiments shown in Figure 3C need to be better described in the text. The rationale for analyzing the genes assessed in Figure 3D need to be clearly stated.
A2.10. We thank the reviewer for this observation. As suggested, we have rewritten the Section 3.3 in our revised manuscript. Please kindly see the line 339-377.
3.3. MALAT1 expression modulates HCC oncogenicity and stemness via interaction with Wnt/β-catenin
Having demonstrated that MALAT1 expression is associated with the progression and recurrence of HCC earlier, against the background of increasing evidence linking cancer progression and recurrence to the presence of CSCs [15, 16], we investigated if and how MALAT1 expression modulates the stem cell-like phenotype of HCC cells using the wild type SK-Hep1 culture cells and MALAT1 knockdown clones #1 and #2. By FACS, compared to 7.2% CSCs-like side population (SP) cells identified in control SK-Hep1 culture cells bearing scrambled shRNA, we observed only 1.8% and 1.1% CSCs-like SP cells in shMALAT1#1 and shMALAT1#2 SK-Hep1 cultures, respectively (Figure 3A, upper). Further, in similar assay, we evaluated the effect of MALAT1 expression on ALDH activity using the Aldefluor™ flow cytometry assay. Side scatter of incident light in control and shMALAT1 SK-Hep1 cells showed that silencing MALAT1 reduced the ALDH1 activity in the shMALAT1#1 and shMALAT1#2 (Figure 3A, lower). These results were validated by Western blot analyses results showing that compared to the scrambled control cells, the expression levels of CD133, ALDH1 and β-catenin proteins in shMALAT1 SK-Hep1 or HepG2 cells were significantly down-regulated (Figure 3B). We also observed significant reduction in the number of invaded shMALAT1#1 (75%, p < 0.001) or shMALAT1#2 (83%, p < 0.001) cells, compared to the scrambled control (Figure 3C, left); similarly, clonogenicity was suppressed in shMALAT1#1 and shMALAT1#2 SK-Hep1 cells, respectively, compared to the scrambled control (Figure 3C, right). In addition, we demonstrated that in MALAT1 knockdown HCC clones, the expression of c-Myc, CK19, β-catenin, p-Stat3, Stat3, vimentin and Twist1 proteins was significantly suppressed, compared to the scrambled control cells (Figure 3D). Observing a correlation in the expression of β-catenin, a principal component of the canonical Wnt signaling pathway, which is implicated in the expansion of CSCs population and induction of epithelial-to-mesenchymal transition (EMT) (17), we sought to understand if the demonstrated oncogenic and stemness activities of MALAT1 were via its direct interaction or interplay with β-catenin. Following sequence-based prediction of interaction between MALAT1 (NCBI Reference Sequence: NR_002819.4) and , β-catenin (NCBI Reference Sequence: NP_001895.1), with 0.85 and 0.97 interaction probabilities using random forest (RF) and support-vector machines classifiers, respectively, we further used the Schrödinger® PyMOL molecular graphics software version 2.3.2 (https://pymol.org/2/) to create a visualization of the molecular interaction between MALAT1 (PDB ID: 4PLX) and β-catenin (PDB ID: 3BCT), thus showing that MALAT1 directly targets and binds to β-catenin with a root-mean-square deviation (RMSD) of 4.0 Å, docking score of 19478, β-catenin/MALAT1 complex interface area of 3077.50 Å2, and an atomic contact energy (ACE) of -298.23 kcal/mol , and that this interaction may bear some functional connotations, especially in the context of HCC metastatic and CSCs-like phenotype (Figure 3E). For additional validation, we probed the effect of immunoprecipitating β-catenin with MALAT1 on the level bound transcripts using the RNA immunoprecipitation (RIP) assay, since this assay allows interrogation of the physical association between RNA binding proteins and transcripts of interest. Our RIP analysis demonstrates that both MALAT1-β-catenin molecules can be co-immunoprecipitated, but other interacting molecules could be involved (Figure 3F).
Q2.11. Figure 3A upper panel/lines 339: the figures in the panels and the text are different; this is confusing. The size of the panels could be increased. The labeling appears to be blurred.
A2.11. We thank the reviewer for this observation. As suggested, we have now included the statistical analysis in newly Figure 3A in our revised manuscript. Futhermore, we have rewritten the Section 3.3 in our revised manuscript. Please kindly see the line 339-377.
3.3. MALAT1 expression modulates HCC oncogenicity and stemness via interaction with Wnt/β-catenin
Having demonstrated that MALAT1 expression is associated with the progression and recurrence of HCC earlier, against the background of increasing evidence linking cancer progression and recurrence to the presence of CSCs [15, 16], we investigated if and how MALAT1 expression modulates the stem cell-like phenotype of HCC cells using the wild type SK-Hep1 culture cells and MALAT1 knockdown clones #1 and #2. By FACS, compared to 7.2% CSCs-like side population (SP) cells identified in control SK-Hep1 culture cells bearing scrambled shRNA, we observed only 1.8% and 1.1% CSCs-like SP cells in shMALAT1#1 and shMALAT1#2 SK-Hep1 cultures, respectively (Figure 3A, upper). Further, in similar assay, we evaluated the effect of MALAT1 expression on ALDH activity using the Aldefluor™ flow cytometry assay. Side scatter of incident light in control and shMALAT1 SK-Hep1 cells showed that silencing MALAT1 reduced the ALDH1 activity in the shMALAT1#1 and shMALAT1#2 (Figure 3A, lower). These results were validated by Western blot analyses results showing that compared to the scrambled control cells, the expression levels of CD133, ALDH1 and β-catenin proteins in shMALAT1 SK-Hep1 or HepG2 cells were significantly down-regulated (Figure 3B). We also observed significant reduction in the number of invaded shMALAT1#1 (75%, p < 0.001) or shMALAT1#2 (83%, p < 0.001) cells, compared to the scrambled control (Figure 3C, left); similarly, clonogenicity was suppressed in shMALAT1#1 and shMALAT1#2 SK-Hep1 cells, respectively, compared to the scrambled control (Figure 3C, right). In addition, we demonstrated that in MALAT1 knockdown HCC clones, the expression of c-Myc, CK19, β-catenin, p-Stat3, Stat3, vimentin and Twist1 proteins was significantly suppressed, compared to the scrambled control cells (Figure 3D). Observing a correlation in the expression of β-catenin, a principal component of the canonical Wnt signaling pathway, which is implicated in the expansion of CSCs population and induction of epithelial-to-mesenchymal transition (EMT) (17), we sought to understand if the demonstrated oncogenic and stemness activities of MALAT1 were via its direct interaction or interplay with β-catenin. Following sequence-based prediction of interaction between MALAT1 (NCBI Reference Sequence: NR_002819.4) and , β-catenin (NCBI Reference Sequence: NP_001895.1), with 0.85 and 0.97 interaction probabilities using random forest (RF) and support-vector machines classifiers, respectively, we further used the Schrödinger® PyMOL molecular graphics software version 2.3.2 (https://pymol.org/2/) to create a visualization of the molecular interaction between MALAT1 (PDB ID: 4PLX) and β-catenin (PDB ID: 3BCT), thus showing that MALAT1 directly targets and binds to β-catenin with a root-mean-square deviation (RMSD) of 4.0 Å, docking score of 19478, β-catenin/MALAT1 complex interface area of 3077.50 Å2, and an atomic contact energy (ACE) of -298.23 kcal/mol , and that this interaction may bear some functional connotations, especially in the context of HCC metastatic and CSCs-like phenotype (Figure 3E). For additional validation, we probed the effect of immunoprecipitating β-catenin with MALAT1 on the level bound transcripts using the RNA immunoprecipitation (RIP) assay, since this assay allows interrogation of the physical association between RNA binding proteins and transcripts of interest. Our RIP analysis demonstrates that both MALAT1-β-catenin molecules can be co-immunoprecipitated, but other interacting molecules could be involved (Figure 3F).
Q2.12. Figure 3C: the statistical analysis appears to be incorrectly indicated unless the authors compared both MALAT1 sh clones? The size of the upper panels could be increased.
A2.12. We thank the reviewer for this observation. As suggested, we have now included the statistical analysis in newly Figure 3C in our revised manuscript.
Q2.13. Figure 3F: the authors should more carefully interpret their data. The sentence in lines 370-371 is an overstatement. RIP does not demonstrate a direct interaction between MALAT1 and beta-catenin. This assay demonstrates that both molecules can be coimmunoprecipitated, but other interacting molecules could be involved.
A2.13. We thank the reviewer for this observation. We have rewritten the Section 3.3 in our revised manuscript. Please kindly see the line 339-377.
3.3. MALAT1 expression modulates HCC oncogenicity and stemness via interaction with Wnt/β-catenin
Having demonstrated that MALAT1 expression is associated with the progression and recurrence of HCC earlier, against the background of increasing evidence linking cancer progression and recurrence to the presence of CSCs [15, 16], we investigated if and how MALAT1 expression modulates the stem cell-like phenotype of HCC cells using the wild type SK-Hep1 culture cells and MALAT1 knockdown clones #1 and #2. By FACS, compared to 7.2% CSCs-like side population (SP) cells identified in control SK-Hep1 culture cells bearing scrambled shRNA, we observed only 1.8% and 1.1% CSCs-like SP cells in shMALAT1#1 and shMALAT1#2 SK-Hep1 cultures, respectively (Figure 3A, upper). Further, in similar assay, we evaluated the effect of MALAT1 expression on ALDH activity using the Aldefluor™ flow cytometry assay. Side scatter of incident light in control and shMALAT1 SK-Hep1 cells showed that silencing MALAT1 reduced the ALDH1 activity in the shMALAT1#1 and shMALAT1#2 (Figure 3A, lower). These results were validated by Western blot analyses results showing that compared to the scrambled control cells, the expression levels of CD133, ALDH1 and β-catenin proteins in shMALAT1 SK-Hep1 or HepG2 cells were significantly down-regulated (Figure 3B). We also observed significant reduction in the number of invaded shMALAT1#1 (75%, p < 0.001) or shMALAT1#2 (83%, p < 0.001) cells, compared to the scrambled control (Figure 3C, left); similarly, clonogenicity was suppressed in shMALAT1#1 and shMALAT1#2 SK-Hep1 cells, respectively, compared to the scrambled control (Figure 3C, right). In addition, we demonstrated that in MALAT1 knockdown HCC clones, the expression of c-Myc, CK19, β-catenin, p-Stat3, Stat3, vimentin and Twist1 proteins was significantly suppressed, compared to the scrambled control cells (Figure 3D). Observing a correlation in the expression of β-catenin, a principal component of the canonical Wnt signaling pathway, which is implicated in the expansion of CSCs population and induction of epithelial-to-mesenchymal transition (EMT) (17), we sought to understand if the demonstrated oncogenic and stemness activities of MALAT1 were via its direct interaction or interplay with β-catenin. Following sequence-based prediction of interaction between MALAT1 (NCBI Reference Sequence: NR_002819.4) and , β-catenin (NCBI Reference Sequence: NP_001895.1), with 0.85 and 0.97 interaction probabilities using random forest (RF) and support-vector machines classifiers, respectively, we further used the Schrödinger® PyMOL molecular graphics software version 2.3.2 (https://pymol.org/2/) to create a visualization of the molecular interaction between MALAT1 (PDB ID: 4PLX) and β-catenin (PDB ID: 3BCT), thus showing that MALAT1 directly targets and binds to β-catenin with a root-mean-square deviation (RMSD) of 4.0 Å, docking score of 19478, β-catenin/MALAT1 complex interface area of 3077.50 Å2, and an atomic contact energy (ACE) of -298.23 kcal/mol , and that this interaction may bear some functional connotations, especially in the context of HCC metastatic and CSCs-like phenotype (Figure 3E). For additional validation, we probed the effect of immunoprecipitating β-catenin with MALAT1 on the level bound transcripts using the RNA immunoprecipitation (RIP) assay, since this assay allows interrogation of the physical association between RNA binding proteins and transcripts of interest. Our RIP analysis demonstrates that both MALAT1-β-catenin molecules can be co-immunoprecipitated, but other interacting molecules could be involved (Figure 3F).
Q2.14. Section 3.4: the experiments need to be better described. The conclusion appears to be overstated (not all protein levels appear to be significantly reduced).
A2.14. We thank the reviewer for this observation. As suggested, we have now included the statistical analysis in newly Figure 4 in our revised manuscript. Furthermore, we have rewritten the Section 3.4 in our revised manuscript. Please kindly see the line 394-414.
3.4. MALAT1 oncogenic activity in HCC is mediated by the Wnt/β-catenin signaling pathway
Despite evidence that β-catenin is a principal component of the canonical Wnt signaling pathway and that transcriptional response to Wnt signal activation is coordinated by a complex mesh of factors that bind to β-catenin in the nucleus [18], the mechanisms underlying MALAT1-mediated oncogenicity and CSCs-like phenotypes in HCC cells remain uncoupled with the activation of Wnt/β-catenin signaling. Thus, using the TOP/FOP flash luciferase reporter assay, we investigated the effect of MALAT1 on β-catenin/TCF-dependent transcriptional activity as an indicator of the role of MALAT1 in the modulation of the canonical Wnt signaling pathway, and the probable role this may play in MALAT1-mediated oncogenicity. The TOP/FOP flash reporter activities in shMALAT1#2 SK-Hep1 and HepG2 cells were significantly inhibited (SK-Hep1: 71%, p < 0.01; HepG2, 52%, p < 0.01), compared to the scrambled control cells (Figure 4A). Furthermore, the mRNA expression of β-catenin and its downstream targets, which are effector genes of the Wnt/β-catenin pathway, namely cyclin D1, c-Myc, Axin2, LEF1, and DKK1 in SK-Hep1 and HepG2 cells were evaluated by quantitative PCR. The relative mRNA expression levels of cyclin D1, c-Myc, Axin2, LEF1, and DKK1 were all significantly down-regulated in the MALAT1-silenced SK-Hep1 and HepG2 cells compared to their scrambled control counterparts (Figure 4B). Consistent with the above, western blot analysis showed that the expression of β-catenin, Stat3, c-Myc, cyclin D1, Axin2, LEF1, and DKK1 proteins in the shMALAT1#2 SK-Hep1 and HepG2 cells were significantly reduced, compared to their control scrambled shRNA counterparts (Figure 4C and 4D). These results are indicative of the probable role of MALAT1 in the modulation of the Wnt/β-catenin signaling pathway via direct interaction with β-catenin in the oncogenicity of HCC.
Q2.15. Figure 5B, section 3.5: this needs to be better described. The conclusion does not appear to be in line with the shown data. The authors are mixing data from CDDP treatment and MALAT1 silencing to conclude that "Interestingly, we observed that contrary to conventional knowledge, CDDP increased the population of CD133highCD90high SK-Hep1 429 (28.1%) or HepG2 (22.7%) cells, compared to vehicle-treated scrambled control (Figure 5B, upper), suggesting a CDDP-induced HCC-SCs phenotype. However, this enhanced CD133/90 positivity was markedly repressed in the shMALAT1#1 and shMALAT1#2 HepG2 or SK-Hep1 cells (Figure 5B, lower)." Given the experimental set up the authors cannot link CDDP and MALAT1 silencing data as they did not add CDDP to MALAT1 sh cells. Thus, the conclusion (line 533) does not appear to be supported by the shown data.
A2.15. We thank the reviewer for this observation. As suggested, we have now included the CDDP to MALAT1 sh cells analysis in newly Figure 5 in our revised manuscript. Futhermore, we have rewritten the Section 3.5 in our revised manuscript. Please kindly see the line 424-450.
3.5. Silencing MALAT1 is associated with reduced CD133highCD90high HCC population with suppressed HCC tumorsphere formation in vitro
Having shown that the oncogenic activity of MALAT1 in HCC is associated with enhanced expression of pluripotency markers CD133 and ALDH1, and is mediated, at least in part, by the Wnt/β-catenin signaling pathway, in vitro, we further investigated the translational relevance of these findings in the context of current anticancer chemotherapy, and in in vivo murine HCC models. In corroboratory assays, we demonstrated that concomitantly with reduced tumorsphere size and quantity in the shMALAT1 cells, the immunoreactivity of the cancer-associated pluripotency marker CD133, and mesenchymal and/or liver stem cell marker CD90/THY-1 [19, 20] was significantly suppressed in the shMALAT1 cells, compared to the scrambled control (Figure 5A). We also demonstrated that shMALAT1#1 and shMALAT1#2 markedly inhibited the self-renewal potential of the SK-Hep1 (shMALAT1#1: 82% inhibition, p < 0.001; shMALAT1#2: 92.3% inhibition, p < 0.001) or HepG2 (shMALAT1#1: 83% inhibition, p < 0.05; shMALAT1#2: 93.5% inhibition, p < 0.01) in primary and secondary tumorspheres (Figure 5B). Next, against the background that cisplatin (CDDP)-based hepatic arterial infusion chemotherapy enhances the objective response rate and survival benefit in patients with advanced HCC (19), we examined the probable effect of shMALAT1 on the response to CDDP treatment. Interestingly, we observed that contrary to conventional knowledge, CDDP increased the population of CD133highCD90high SK-Hep1 (28.1%) or HepG2 (22.7%) cells, compared to vehicle-treated scrambled control (Figure 5C), suggesting a CDDP-induced HCC-SCs phenotype. However, this enhanced CD133/90 positivity was markedly repressed in the shMALAT1#1 and shMALAT1#2 HepG2 or SK-Hep1 cells. shMALAT1#2 in combination with CDDP significantly inhibited population of CD133highCD90high across both HepG2 and SK-Hep1 cells (Figure 5C). Furthermore, to gain better insight into the relation between enhanced CD133/CD90 positivity and MALAT1 expression, our re-analysis of the TCGA-LIHC cohort (n = 374) showed that consistent with our earlier results, the expression of CD133/PROM1 is positively correlated with CD90/THY1 (r = 0.359, p = 8.55 x 10−13), and that MALAT1 exhibits positive correlation with CD133/PROM1 (r = 0.165, p = 1.33 x 10−3) and CD90/THY1 (r = 0.165, p = 1.33 x 10−3) (Figure 5D).
Q2.16. To strengthen their conclusions functional rescue experiments should be included.
A2.16. We thank the reviewer for this observation. We will include the functional rescue experiments in our ongoing study.
Q2.17. Figure 5C: "significant positive correlation" is an overstatement.
A2.17. We thank the reviewer for this observation. We have rewritten the Section 3.5 in our revised manuscript. Please kindly see the line 424-450.
3.5. Silencing MALAT1 is associated with reduced CD133highCD90high HCC population with suppressed HCC tumorsphere formation in vitro
Having shown that the oncogenic activity of MALAT1 in HCC is associated with enhanced expression of pluripotency markers CD133 and ALDH1, and is mediated, at least in part, by the Wnt/β-catenin signaling pathway, in vitro, we further investigated the translational relevance of these findings in the context of current anticancer chemotherapy, and in in vivo murine HCC models. In corroboratory assays, we demonstrated that concomitantly with reduced tumorsphere size and quantity in the shMALAT1 cells, the immunoreactivity of the cancer-associated pluripotency marker CD133, and mesenchymal and/or liver stem cell marker CD90/THY-1 [19, 20] was significantly suppressed in the shMALAT1 cells, compared to the scrambled control (Figure 5A). We also demonstrated that shMALAT1#1 and shMALAT1#2 markedly inhibited the self-renewal potential of the SK-Hep1 (shMALAT1#1: 82% inhibition, p < 0.001; shMALAT1#2: 92.3% inhibition, p < 0.001) or HepG2 (shMALAT1#1: 83% inhibition, p < 0.05; shMALAT1#2: 93.5% inhibition, p < 0.01) in primary and secondary tumorspheres (Figure 5B). Next, against the background that cisplatin (CDDP)-based hepatic arterial infusion chemotherapy enhances the objective response rate and survival benefit in patients with advanced HCC (19), we examined the probable effect of shMALAT1 on the response to CDDP treatment. Interestingly, we observed that contrary to conventional knowledge, CDDP increased the population of CD133highCD90high SK-Hep1 (28.1%) or HepG2 (22.7%) cells, compared to vehicle-treated scrambled control (Figure 5C), suggesting a CDDP-induced HCC-SCs phenotype. However, this enhanced CD133/90 positivity was markedly repressed in the shMALAT1#1 and shMALAT1#2 HepG2 or SK-Hep1 cells. shMALAT1#2 in combination with CDDP significantly inhibited population of CD133highCD90high across both HepG2 and SK-Hep1 cells (Figure 5C). Furthermore, to gain better insight into the relation between enhanced CD133/CD90 positivity and MALAT1 expression, our re-analysis of the TCGA-LIHC cohort (n = 374) showed that consistent with our earlier results, the expression of CD133/PROM1 is positively correlated with CD90/THY1 (r = 0.359, p = 8.55 x 10−13), and that MALAT1 exhibits positive correlation with CD133/PROM1 (r = 0.165, p = 1.33 x 10−3) and CD90/THY1 (r = 0.165, p = 1.33 x 10−3) (Figure 5D).
Q2.18. General comment on figures to increase their readability: FACS panels should be improved (increase size, provide clear labeling). The molecular weight of the proteins detected by Western blot should be shown.
A2.18. We thank the reviewer for this observation. The molecular weight of the proteins was added on all of the western blot in our revised manuscript.

Reviewer 3 Report
The authors found that MALAT1 is highly expressed in HCC and is associated with aggressive phenotype. They investigated the role of MALAT1 in the stem-cells-like phenotype and found that MALAT1 attenuates HCC tumorsphere formation efficiency with reduction in CD133+ and CD90+ HCC cell population. The authors suggest that therapeutic targeting of MALAT1/Wnt may constitute a novel promising anticancer strategy for HCC treatment. Targeting MALAT1/Wnt signaling as a therapeutic approach is interesting. The following concerns need attention:
- It is not clear how the shMALAT1 HCC cells were generated. Please provide information on how they were generated.
- Self-renew is unique ability of stem cells and distinguishes them from other cells type. On figure 5A, the authors used first generation of tumorspheres on the experiments. (line 195: 5-7 days followed by visualization of tumorspheres…). The authors should perform experiments with at least 3rd and 4th generation tumorspheres from their shMALAT1 HCC cells (question 1).
- Figure 5B, the authors did not add cisplatin in shMALAT1 cells, so they can not draw any conclusion regarding MALAT1, cisplatin and stemness. The effect of MALAT1 on stemness is not related with cisplatin. The authors should include cisplatin on their experiments with shMALAT1 cells.
- On Material and Methods session, description of invasion is on migration session. Please check lines168-183.
- For western blot, how did the authors quantify the bands?
- There are some missing spaces and typo errors throughout the manuscript.
Author Response
Point-by-point responses to reviewer’s comments - Reviewer 3:
We would like to thank the reviewer for the thorough reading of our manuscript as well as their valuable comments. We believe all comments are borne out of good faith, and thus, have tried to address their comments conscientiously and feel that they have further improved the readability and appeal of our work, as well as strengthened the manuscript. Below are our point-by-point responses.
Q3.1. The authors found that MALAT1 is highly expressed in HCC and is associated with aggressive phenotype. They investigated the role of MALAT1 in the stem-cells-like phenotype and found that MALAT1 attenuates HCC tumorsphere formation efficiency with reduction in CD133+ and CD90+ HCC cell population. The authors suggest that therapeutic targeting of MALAT1/Wnt may constitute a novel promising anticancer strategy for HCC treatment. Targeting MALAT1/Wnt signaling as a therapeutic approach is interesting. The following concerns need attention
A3.1. We are very grateful to the reviewer for taking time to read our manuscript and for the critiques and suggestions made in order to help us improve the quality of our work. In this revised manuscript, we have made effort to address all the comments and suggestions.
Q3.2. It is not clear how the shMALAT1 HCC cells were generated. Please provide information on how they were generated.
A3.2. We sincerely appreciate the reviewer’s comment. We have rewritten the 2.3. MALAT1 silencing in our revised manuscript. Please kindly see our revised Material and Methods section, Lines 115-124.
2.3. MALAT1 silencing
The shRNA specifically targeting MALAT1 was constructed by CRISPR Gene Targeting Core Lab (Taipei Medical University, Taiwan). For MALAT1 silencing, SK-Hep1 and HepG2 cells were transfected with MALAT1 shRNA (shRNA#1: Forward 5’-CCGGAAAGCCCTGA ACTATCACACTCTCGAGAGTGTGATAGTTCAGGGCTTTTTTG-3’; Reverse 5’-AATTCAAAAAAAAGCCCTGAACTATCACACTCTCGAGAGTGT GATAGTTCAGGGCTTT -3’ or shRNA#2: Forward 5’- CCGGAATCTGTAAGCAGTTT GTATGCTCGAGCATACAAA CTGCTTACAGATTTTTTTG-3’; Reverse 5’- AATTCAAAAAAA TCTGTAAGCAGTTTGTATGCT CGAGCATACAAACTGCTTACAGATT -3’) or vector, then the stably transfected shMALAT SK-Hep1 and HepG2 cells were selected with 1μg/ml puromycin.
Q3.3. Self-renew is unique ability of stem cells and distinguishes them from other cells type. On figure 5A, the authors used first generation of tumorspheres on the experiments. (line 195: 5-7 days followed by visualization of tumorspheres…). The authors should perform experiments with at least 3rd and 4th generation tumorspheres from their shMALAT1 HCC cells (question 1).
A3.3. We thank the reviewer for this observation. As suggested, we have now included the primary and secondary tumorspheres analysis in newly Figure 5 in our revised manuscript. Furthermore, we have rewritten the Section 3.5 in our revised manuscript. Please kindly see the line 424-450.
3.5. Silencing MALAT1 is associated with reduced CD133highCD90high HCC population with suppressed HCC tumorsphere formation in vitro
Having shown that the oncogenic activity of MALAT1 in HCC is associated with enhanced expression of pluripotency markers CD133 and ALDH1, and is mediated, at least in part, by the Wnt/β-catenin signaling pathway, in vitro, we further investigated the translational relevance of these findings in the context of current anticancer chemotherapy, and in in vivo murine HCC models. In corroboratory assays, we demonstrated that concomitantly with reduced tumorsphere size and quantity in the shMALAT1 cells, the immunoreactivity of the cancer-associated pluripotency marker CD133, and mesenchymal and/or liver stem cell marker CD90/THY-1 [19, 20] was significantly suppressed in the shMALAT1 cells, compared to the scrambled control (Figure 5A). We also demonstrated that shMALAT1#1 and shMALAT1#2 markedly inhibited the self-renewal potential of the SK-Hep1 (shMALAT1#1: 82% inhibition, p < 0.001; shMALAT1#2: 92.3% inhibition, p < 0.001) or HepG2 (shMALAT1#1: 83% inhibition, p < 0.05; shMALAT1#2: 93.5% inhibition, p < 0.01) in primary and secondary tumorspheres (Figure 5B). Next, against the background that cisplatin (CDDP)-based hepatic arterial infusion chemotherapy enhances the objective response rate and survival benefit in patients with advanced HCC (19), we examined the probable effect of shMALAT1 on the response to CDDP treatment. Interestingly, we observed that contrary to conventional knowledge, CDDP increased the population of CD133highCD90high SK-Hep1 (28.1%) or HepG2 (22.7%) cells, compared to vehicle-treated scrambled control (Figure 5C), suggesting a CDDP-induced HCC-SCs phenotype. However, this enhanced CD133/90 positivity was markedly repressed in the shMALAT1#1 and shMALAT1#2 HepG2 or SK-Hep1 cells. shMALAT1#2 in combination with CDDP significantly inhibited population of CD133highCD90high across both HepG2 and SK-Hep1 cells (Figure 5C). Furthermore, to gain better insight into the relation between enhanced CD133/CD90 positivity and MALAT1 expression, our re-analysis of the TCGA-LIHC cohort (n = 374) showed that consistent with our earlier results, the expression of CD133/PROM1 is positively correlated with CD90/THY1 (r = 0.359, p = 8.55 x 10−13), and that MALAT1 exhibits positive correlation with CD133/PROM1 (r = 0.165, p = 1.33 x 10−3) and CD90/THY1 (r = 0.165, p = 1.33 x 10−3) (Figure 5D).
Please kindly see the newly Figure 5 legend.
Figure 5. Silencing MALAT1 is associated with reduced CD133highCD90high HCC population with suppressed HCC tumorsphere formation in vitro. (A) Photo-images showing the effect of shMALAT1 #1 and #2 on the size of tumorspheres formed and on the expression of CD133 and CD90 (left panel). Histograms show the effect of shMALAT1 #1 and #2 on number of tumorspheres formed (right panel). (B) SK-Hep1 or HepG2 cells transfected with shMALAT1 #1 and #2 exhibited decreased HCC tumorsphere size and number in both primary and secondary generation tumorsphere. (C) Comparative flow-cytometry analysis CD133 and CD90 positivity in SK-Hep1 or HepG2 cells exposed to CDDP or transfected with shMALAT1 #1 or #2. (D) Graphical representation of the correlation between MALAT1, CD133, and CD90. CDDP, cisplatin; **p< 0.01, ***p< 0.001.
Q3.4. Figure 5B, the authors did not add cisplatin in shMALAT1 cells, so they can not draw any conclusion regarding MALAT1, cisplatin and stemness. The effect of MALAT1 on stemness is not related with cisplatin. The authors should include cisplatin on their experiments with shMALAT1 cells.
A3.4. We thank the reviewer for this observation. As suggested, we have now included the CDDP to MALAT1 sh cells analysis in newly Figure 5 in our revised manuscript. Futhermore, we have rewritten the Section 3.5 in our revised manuscript. Please kindly see the line 424-450.
3.5. Silencing MALAT1 is associated with reduced CD133highCD90high HCC population with suppressed HCC tumorsphere formation in vitro
Having shown that the oncogenic activity of MALAT1 in HCC is associated with enhanced expression of pluripotency markers CD133 and ALDH1, and is mediated, at least in part, by the Wnt/β-catenin signaling pathway, in vitro, we further investigated the translational relevance of these findings in the context of current anticancer chemotherapy, and in in vivo murine HCC models. In corroboratory assays, we demonstrated that concomitantly with reduced tumorsphere size and quantity in the shMALAT1 cells, the immunoreactivity of the cancer-associated pluripotency marker CD133, and mesenchymal and/or liver stem cell marker CD90/THY-1 [19, 20] was significantly suppressed in the shMALAT1 cells, compared to the scrambled control (Figure 5A). We also demonstrated that shMALAT1#1 and shMALAT1#2 markedly inhibited the self-renewal potential of the SK-Hep1 (shMALAT1#1: 82% inhibition, p < 0.001; shMALAT1#2: 92.3% inhibition, p < 0.001) or HepG2 (shMALAT1#1: 83% inhibition, p < 0.05; shMALAT1#2: 93.5% inhibition, p < 0.01) in primary and secondary tumorspheres (Figure 5B). Next, against the background that cisplatin (CDDP)-based hepatic arterial infusion chemotherapy enhances the objective response rate and survival benefit in patients with advanced HCC (19), we examined the probable effect of shMALAT1 on the response to CDDP treatment. Interestingly, we observed that contrary to conventional knowledge, CDDP increased the population of CD133highCD90high SK-Hep1 (28.1%) or HepG2 (22.7%) cells, compared to vehicle-treated scrambled control (Figure 5C), suggesting a CDDP-induced HCC-SCs phenotype. However, this enhanced CD133/90 positivity was markedly repressed in the shMALAT1#1 and shMALAT1#2 HepG2 or SK-Hep1 cells. shMALAT1#2 in combination with CDDP significantly inhibited population of CD133highCD90high across both HepG2 and SK-Hep1 cells (Figure 5C). Furthermore, to gain better insight into the relation between enhanced CD133/CD90 positivity and MALAT1 expression, our re-analysis of the TCGA-LIHC cohort (n = 374) showed that consistent with our earlier results, the expression of CD133/PROM1 is positively correlated with CD90/THY1 (r = 0.359, p = 8.55 x 10−13), and that MALAT1 exhibits positive correlation with CD133/PROM1 (r = 0.165, p = 1.33 x 10−3) and CD90/THY1 (r = 0.165, p = 1.33 x 10−3) (Figure 5D).
Q3.5. On Material and Methods session, description of invasion is on migration session. Please check lines168-183.
A3.5. We sincerely appreciate the reviewer’s comment. We have rewritten the 2.7. Matrigel invasion assay in our revised manuscript. Please kindly see our revised Material and Methods section, Lines 176-184.
2.7. Matrigel invasion assay
For analysis of the invasion potential of scrambled shRNA or shMALAT1 HCC cells, the 24-well plate Transwell system was used. 3.5 × 104 cells were seeded onto the upper chambers of the inserts (BD Bioscience, 8 μm pore size) containing serum-free media, while the lower chambers contained media with 10 % FBS serving as chemo-attractant. Medium was discarded after 24 h incubation, and then non-invaded cells remaining on the upper side of the inserts were removed with sterile cotton swabs while invaded cells underneath the filter membranes were fixed with 3.7 % formaldehyde for 1 h and stained with crystal violet. The invaded cells were visualized and evaluated under microscope.
Q3.6. For western blot, how did the authors quantify the bands?
A3.6. We sincerely appreciate the reviewer’s comment. We have quantify the bands and rewritten the 2.4. Western blot analysis assay in our revised manuscript. Please kindly see our revised Material and Methods section, Lines 125-145.
2.4. Western blot analysis
After washing cells with PBS twice and lysing with ice-cold RIPA lysis buffer (#20-188, Sigma-Aldrich Co., St. Louis, MO, USA), the total protein lysate from the wild-type or shMALAT1 HCC cells was centrifuged and pellet collected. Protein concentration was then quantified using Pierce™ BCA protein assay kit (#23227, Thermo Fisher Scientific Inc., Waltham, MA, USA). Equal amount of protein lysate from each sample was run on 10% sodium dodecyl sulfate polyacrylamide gel electrophoresis (SDS-PAGE) then protein transferred to Polyvinylidene difluoride (PVDF) membranes which were blocked in 1X PBS containing 5% skimmed milk, and then incubated with the specific primary antibodies against β-catenin (1:1000, β-Catenin (6B3) Rabbit mAb #9582P, Sigma), CD133 (1:1000, #MAB4310, Sigma), ALDH-1 (1:1000, (D4R9V) Rabbit mAb #12035, Sigma), c-Myc (1:1000, c-Myc Antibody (9E10) (sc-40), Sigma), CK19 (1:1000, monoclonal antibody SAB3300019, Sigma), Stat3 (1:1000, Stat3 (79D7) Rabbit mAb #4904, Sigma), vimentin (1:1000, Anti-Vimentin antibody (ab137321), Sigma), Twist1 (1:1000, Twist (Twist2C1a) Antibody sc-81417, Sigma), cyclin D1 (1:1000, Cyclin D1 Antibody (A-12) (sc-8396), Sigma), Axin2 (1:1000, Axin2 Antibody SAB3500619, Sigma), LEF1 (1:1000, LEF1 (C12A5) Rabbit mAb #2230, Sigma), or DKK1 (1:1000, DKK1 Antibody #4687, Sigma) at 4 °C overnight for protein detection in Supplementary Table S1. After overnight probing, protein bands were identified using anti-mouse or anti-rabbit horseradish peroxidase (HRP)-linked secondary antibody at room temperature for 1 h. The signal was detected using UVP® imaging system (Analytik Jena US LLC., Upland, CA, USA). β-actin (1:10000, 8H10D10, Mouse mAb #3700) was used as loading control. The gray value was quantified and analyzed using Image J software. The experiment was repeated 3 times.
Q3.7. There are some missing spaces and typo errors throughout the manuscript.
A3.7. We are very grateful to the reviewer for taking time to read our manuscript and for the critiques and suggestions made in order to help us improve the quality of our work. In this revised manuscript, we have made effort to correct all the missing spaces and typo errors throughout the manuscript.

Round 2
Reviewer 2 Report
General comment
This is a revised version of the manuscript previously submitted by Chang et al. Globally the authors have addressed the majority of the previously raised comments although some questions remain as indicated below.
Specific comments
- Lines 115-124: it is not yet clear how MALAT1 silencing was done. This paragraph states "transfection of shRNA or vector". Were lentiviral particles carrying the shRNA produced and used to transduce the cells? If not, what does "vector" refer to? This needs to be clarified. Neither in the material and methods nor in the results section is included a statement about the potency of MALAT1 silencing in these cells, i.e. how much MALAT1 these clones express as compared to the parental cells. In the absence of the characterization of these clones and rescue experiments, it is difficult to exclude an off-targeting effect of the shRNA. If the authors perform rescue experiments, why don't they include them in the revised manuscript and indicate "We will include the functional rescue experiments in our ongoing study"?
- Line 254-256: please double-check the sentence for accuracy "Post-experiment, all animals were humanely sacrificed by tumor dislocation and the tumor samples were harvested for further analyses." cervical dislocation?
- Figure 3A: the FACS panels are still very small and difficult to read. There seems to be still some space in the figure to increase the size of these panels.
- Figure 4C and D: the frames around the western blots are not well aligned with the images.
Reviewer 3 Report
The manuscript has improved. Two minor comments:
- Matrigel invasion assay: the authors performed migration (not invasion) as they did not add Matrigel on the assay. The authors should write Migration assay on Material and Methods and refer as migrated cells (not invaded cells).
- Tumorsphere formation assays: it will be nice if the authors describe how they passaged Primary tumorspheres to Secondary tumorspheres.
